# Robust manipulation of the behavior of *Drosophila melanogaster* by a fungal pathogen in the laboratory

Carolyn Elya[1‡*], Tin Ching Lok[1†], Quinn E Spencer[1†], Hayley McCausland[1], Ciera C Martinez[1], Michael Eisen[1,2,3*]

[1]Department of Molecular and Cell Biology, University of California, Berkeley, Berkeley, United States; [2]Department of Integrative Biology, University of California, Berkeley, Berkeley, United States; [3]Howard Hughes Medical Institute, University of California, Berkeley, Berkeley, United States

**\*For correspondence:**
cnelya@gmail.com (CE);
mbeisen@berkeley.edu (ME)

[†]These authors contributed equally to this work

**Present address:** [‡]Department of Organismic and Evolutionary Biology, Harvard University, Cambridge, United States

**Competing interests:** The authors declare that no competing interests exist.

**Abstract** Many microbes induce striking behavioral changes in their animal hosts, but how they achieve this is poorly understood, especially at the molecular level. Mechanistic understanding has been largely constrained by the lack of an experimental system amenable to molecular manipulation. We recently discovered a strain of the behavior-manipulating fungal pathogen *Entomophthora muscae* infecting wild *Drosophila*, and established methods to infect *D. melanogaster* in the lab. Lab-infected flies manifest the moribund behaviors characteristic of *E. muscae* infection: hours before death, they climb upward, extend their proboscides, affixing in place, then raise their wings, clearing a path for infectious spores to launch from their abdomens. We found that *E. muscae* invades the nervous system, suggesting a direct means by which the fungus could induce behavioral changes. Given the vast molecular toolkit available for *D. melanogaster*, we believe this new system will enable rapid progress in understanding how *E. muscae* manipulates host behavior.
DOI: https://doi.org/10.7554/eLife.34414.001

## Introduction

Among the most extraordinary products of evolution are microorganisms that can manipulate animal behavior to their advantage. Some of the best characterized examples of this phenomenon are found within the entomopathogenic fungi, fungal species that parasitize insect hosts. The ability to precisely manipulate insect behavior has evolved independently multiple times within the fungal kingdom and the hosts of these fungi span many insect orders (*Araújo and Hughes, 2016*). While some behaviorally-manipulating entomopathogenic fungi have been known to science since the mid nineteenth century, others are only now being discovered and described (*Steinkraus et al., 2017*; *Hodge et al., 2017*; *Cohn, 1855*; *Hughes et al., 2011*; *Arthur, 1886*; *Humber, 1976*). Despite substantial progress made in understanding the ecology and phylogeny of these organisms, the molecular mechanisms by which entomopathogenic fungi hijack the animal nervous system have remained elusive.

Recent molecular and genomic studies of the ant parasite *Ophiocordyceps unilateralis* highlight the opportunities and challenges in studying entomopathogenic fungi of non-model organism hosts. This so-called 'zombie ant fungus' induces an infected ant to climb to a high place and affix itself in place using its mandibles before succumbing to the infection, whereupon the ascomycete devours the insect's tissue as a fruiting body emerges from behind the ant's head and pronotum.

Metabolomic analysis of host brains cultured ex vivo in the presence of *O. unilateralis* has found that the fungus secretes specific sets of metabolites in response to being grown with ex vivo

**eLife digest** We are surrounded by billions of microbes, many of which live on or even inside us. With some of these tiny creatures we share a mutually beneficial relationship, and they help to protect and nourish us in return for food and shelter. But some microbes have taken this relationship a step further and evolved the ability to manipulate the behavior of their host animals to their advantage.

For example, certain fungi can hijack the nervous systems of ants, turning them into 'zombies' that are forced to climb to the precise height of a plant with the right conditions for the fungus to grow. Another fungus, called *Entomophthora muscae,* induces a similar behavior in flies. Like with the ants, this fungus forces the flies to climb to a high location and to attach themselves to the plant with their mouthparts, so providing a basis for the fungus to grow.

Due to the lack of knowledge and tools that allow us to study the genes and cells of many of these host species, it remains unclear how these microbes can manipulate other animals and change their behavior. Now, Elya et al. wanted to find out whether this process can be examined in a well-known 'model organism', the fruit fly.

The fruit flies were infected with *E. muscae* obtained from the wild and activity patterns in the genes of both organisms were tracked. One day after infection with the fungus, the flies developed a strong immune response to the invader, and many genes involved in the immune system were overactive. After two days, the fungus started to adopt the shape that eventually kills the flies and began to invade the nervous system. By the third day, the fungus had spread throughout the body and most flies died around four to five days following infection. Two to three hours after the flies had died, the digestive and reproductive systems had been consumed by the fungus and the nervous system was in the process of being degraded.

Contrary to other infected organisms, the fungus got inside the fly's nervous system during the early stages of infection. This could be an important clue as to how the fungus changes the fly's behavior, but more research is needed to confirm this. The techniques developed in the study of Elya et al. can now be used to study how exactly microbes affect their hosts. A better knowledge of the relationship between these organisms will provide new insight into how animal behaviors are encoded and generated.

DOI: https://doi.org/10.7554/eLife.34414.002

dissected brains from its natural hosts (*de Bekker et al., 2014*). Transcriptomic measurements of the heads of infected ants during and after behavior manipulation revealed changes in the expression of fungal genes responsible for secreted factors and enterotoxins, while changes in ant gene expression indicate a dampened immune response during biting (*de Bekker et al., 2015*). Most recently, microscopy data has shown that *O. unilateralis* does not invade host neural tissue and instead, forms complex fungal networks around host mandibular muscle, which could reflect intercellular communication within the fungus inside the ant body cavity (*Fredericksen et al., 2017*). While this work has helped elucidate the molecular basis of behavioral manipulation, further progress with limited by the lack of established genetic tools in either the fungus or the host and neurobiological tools in the host.

Intriguingly, a similar suite of behaviors are induced in dipterans by the distantly related Zygomycete fungi within the *Entomophthora muscae* species complex (*Keller, 1984*; *MacLeod et al., 1976*). *E. muscae,* whose genus name *Entomophthora* comes from the Greek 'entomo' for insect and 'phthora' for destroyer, was first described in house flies (*Musca domestica*) in 1855 (*Cohn, 1855*). A fly infected with *E. muscae* exhibits a characteristic set of fungus-induced behaviors: shortly before sunset on its final day of life, the fly climbs to a high location (a behavior known as 'summiting'), extends its proboscis and becomes affixed to the surface via fungal holdfasts, hyphal growths that emerge through the mouthparts (*Roy et al., 2006*; *Balazy, 1984*; *Krasnoff et al., 1995*). The fly's wings then lift up and away from its dorsal abdomen, striking a final death pose that is thought to be ideal for fungal dispersal (*Krasnoff et al., 1995*).

Over the course of the next few hours, the fungus differentiates into structures called conidiophores within the dead fly. Conidiophores emerge through the weakest points in the fly's cuticle,

typically the intersegmental membranes of the dorsal abdomen, giving the cadavers a distinct banding pattern (*Brobyn and Wilding, 1983*). A primary conidium (also referred to as a 'spore') forms at the tip of each conidiophore; once mature, these conidia are forcibly ejected into the surrounding environment in order to land on the cuticle of a susceptible fly host (*Mullens and Rodriguez, 1985*; *Mullens et al., 1987*).

Launched primary conidia are polynucleated, campanulate (bell-shaped) and are surrounded by a sticky 'halo' that serves to adhere the conidium where it lands. If successful in landing on the cuticle of a new host, the conidium germinates, using both mechanical and enzymatic force to bore through the cuticle and into the fly's hemolymph (*Brobyn and Wilding, 1983*; *Charnley, 1984*). If the primary conidium misses its target or fails to germinate upon landing on the host (*Watson and Petersen, 1993*), it can sporulate anew to generate a smaller secondary conidium (*Mullens and Rodriguez, 1985*). Off-target conidia can continue to re-sporulate and give rise to smaller, higher order conidia until a host is reached or resources are exhausted (*Mullens and Rodriguez, 1985*). Once inside the fly, the fungus is initially sustained by nutrients in the hemolymph and then later consumes the fat body as an energy source (*Brobyn and Wilding, 1983*). When available resources are depleted, the fungus elicits the previously-described end-of-life behaviors and the fungal life cycle begins again.

The *E. muscae* species complex consists of several genetically, though not always morphologically, distinct species that are capable of infecting dipterans in a variety of fly families (most frequently Muscidae) (*Keller, 2002*, *2007*). In the wild, species within the *E. muscae* species complex have been observed to infect more than one host species (*Mullens et al., 1987*; *Klingen et al., 2000*). However, recent molecular evidence suggests that there may be additional host specialization within some *E. muscae* complex species (*Gryganskyi et al., 2013*; *Jensen et al., 2009*). In the laboratory, a range of fly species can be infected and killed by *E. muscae*, though not all infected hosts manifest the stereotypical end-of-life behaviors, and susceptibility has not been found to track with host phylogeny (*Steinkraus and Kramer, 1987*; *Kramer and Steinkraus, 1981*). *E. muscae* has almost exclusively been observed and studied in muscoid flies (especially the house fly, *Musca domestica*), organisms for which very few experimental tools exist (*Keller, 2007*). Thus, despite inspiring curiosity and intrigue for over a century, how *E. muscae* achieves control of its host remains poorly understood, with essentially no information as to what is occurring at the molecular level in either fungus or host.

Although there have been only a few reports of *E. muscae* infection in wild *Drosophila* (*Keller, 2002*; *Turian and Wuest, 1969*; *Goldstein, 1927*), we observed several wild *Drosophila* in separate sites in Northern California with the characteristic death pose and fungal banding of *E. muscae* infections, and saw an opportunity to study a behavior-changing fungus in the laboratory species, *Drosophila melanogaster*. Here, we describe the isolation and subsequent characterization of this *E. muscae* strain and its impact on *D. melanogaster* behavior in the laboratory, and present the *E. muscae*-*D. melanogaster* system as a model for developing a mechanistic understanding of parasitic manipulation of host behavior.

## Results

### Discovery and isolation of *E. muscae* from wild *Drosophila*

We began collecting wild *Drosophila* in a several field sites in Northern California in fall 2014 as part of several unrelated projects (*Quan and Eisen, 2018*; *Elya et al., 2017*), and noticed several individuals of multiple *Drosophila* species that appeared to have been killed by *E. muscae* (*Supplementary file 1*). Although these flies had all ejected most or all of their spores, we were able to confirm the identity of the fungus by genotyping cadavers at the ITS and 28S (LSU) rDNA loci (*Figure 1—figure supplement 1B,C*).

In June 2015, we established a stable food source (organic fruits in a clean dish pan, referred to henceforth as the 'fendel') at a field site in Berkeley, CA to collect wild *Drosophila* for a separate study (see [*Elya et al., 2016*]). In late July 2015, we noticed that several flies had died at this site with raised wings at the bottom of the fendel and, upon closer inspection, observed remnants of fungal growth and sporulation on these dead flies (*Figure 1—figure supplement 1A*). We suspected that these animals had been killed by the fungal pathogen *Entomophthora muscae*, though there

have been only a few reports of *E. muscae* infection in wild *Drosophila* (*Keller, 2002*; *Turian and Wuest, 1969*; *Goldstein, 1927*).

We confirmed that these flies had been killed by *E. muscae* by genotyping a dozen representative cadavers (*Figure 1—figure supplement 1B,C*). PCR genotyping of the host at the cytochrome oxidase II (COII) locus (*Liu and Beckenbach, 1992*) demonstrated that susceptible host species included *D. melanogaster, D. immigrans, D. simulans* and *D. hydei*, which are all commonly observed in Berkeley, CA. The fungal sequences for all of the typed cadavers at this site and others were identical at these two loci, consistent with one *E. muscae* strain being responsible for this epizootic event.

Species identification within the *E. muscae* species complex (which will hereafter be referred to as *E. muscae*) has historically relied on conidial morphology (and, to a lesser extent, host species), but is expanding to include molecular data (*Hajek et al., 2012*). Still, the taxonomic boundaries between strains and species within this group are still unclear. To distinguish our strain (or possibly species) from others reported, we will henceforth refer to our isolate as *E. muscae* 'Berkeley'.

We were intrigued by the possibility that the presence of *E. muscae* in *Drosophila* would allow us to establish an infection in lab-reared flies. However, our initial observations were all of dead flies that had already ejected their spores (*Figure 1—figure supplement 1A*). Studies in *M. domestica* have shown that, at room temperature, the majority of *E. muscae*'s infectious spores are ejected within the first approximately twelve hours of an infected host's death, and lose infectivity within 48 hr of landing on a non-host substrate (*Kalsbeek et al., 2001*). Thus, to culture *E. muscae* 'Berkeley' we needed to procure freshly-killed flies to ensure access to viable conidia.

The repeated observation of *E. muscae* 'Berkeley'-killed *Drosophila* demonstrated that the infection was circulating in the population of flies at our field site. We therefore reasoned that some of the flies that were visiting our fendel should be infected. Previous *E. muscae* research had demonstrated that the fungus only kills hosts once a day, around sunset (*Krasnoff et al., 1995*). Thus, we collected flies once every morning (1–2 hr after sunrise) from the fendel and monitored them nightly (1–3 hr after sunset), looking for animals that had recently died in the stereotyped death pose (*Figure 1*).

Using a single, wild *D. hydei* cadaver, we first established a culture of *E. muscae* 'Berkeley' in vitro, by inoculating liquid media previously reported to support *E. muscae* growth (*Hajek et al., 2012*). Genotyping the resultant culture at both the ITS and 28S loci verified that we had isolated the same strain as the one that had killed the previously observed cadavers (*Figure 1—figure supplement 2*).

To establish an in vivo infection, wild cadavers were co-housed overnight in a confined space with healthy, lab-reared CantonS *D. melanogaster*, and exposed flies were monitored nightly for two weeks to identify *E. muscae* 'Berkeley' cadavers. We repeated this process daily for several weeks before we were able to passage the infection. For the first few weeks, we housed exposed flies on standard fly diet, which contained a small amount of the preservative tegosept (0.09%). We did not anticipate that this would be problematic since infected wild flies still died of infection after being housed on this diet for up to eight days (*Figure 1—figure supplement 3*). However, it was only when we began housing flies on food devoid of the preservative tegosept that we were able to successfully passage the infection to lab-reared flies.

Once we had transferred *E. muscae* 'Berkeley' to lab-reared flies, we assessed the impacts of several variables on infection efficacy, ultimately arriving at an optimized propagation protocol (*Figure 2—figure supplement 1*). Briefly, *E. muscae* infection is passaged to new flies by embedding six freshly-killed, infected cadavers headfirst in sucrose agar and confining 50 young (eclosed within the past 24 hr) CantonS adults of mixed sex with these cadavers for 24 hr in a cool, humid environment (18°C with saturating humidity) on an inverted 12:12 light:dark cycle. At 1 day (24 hr) post-exposure, confinement is relieved and flies are transferred to a slightly warmer temperature (21°C) and lower humidity (60%). At 2 days (48 hr post-exposure), flies are transferred to fresh medium that lacks tegosept but includes a small quantity of propionic acid to stymie bacterial growth. At 3–7 days (72–144 hr) post-exposure, flies are monitored daily for death by fungus.

## Description of *E. muscae* 'Berkeley' infection in CantonS flies

With *E. muscae* 'Berkeley' stably propagating in vivo, we next focused on carefully observing the process of infection in CantonS flies. By eye, infected flies are hard to distinguish from their healthy

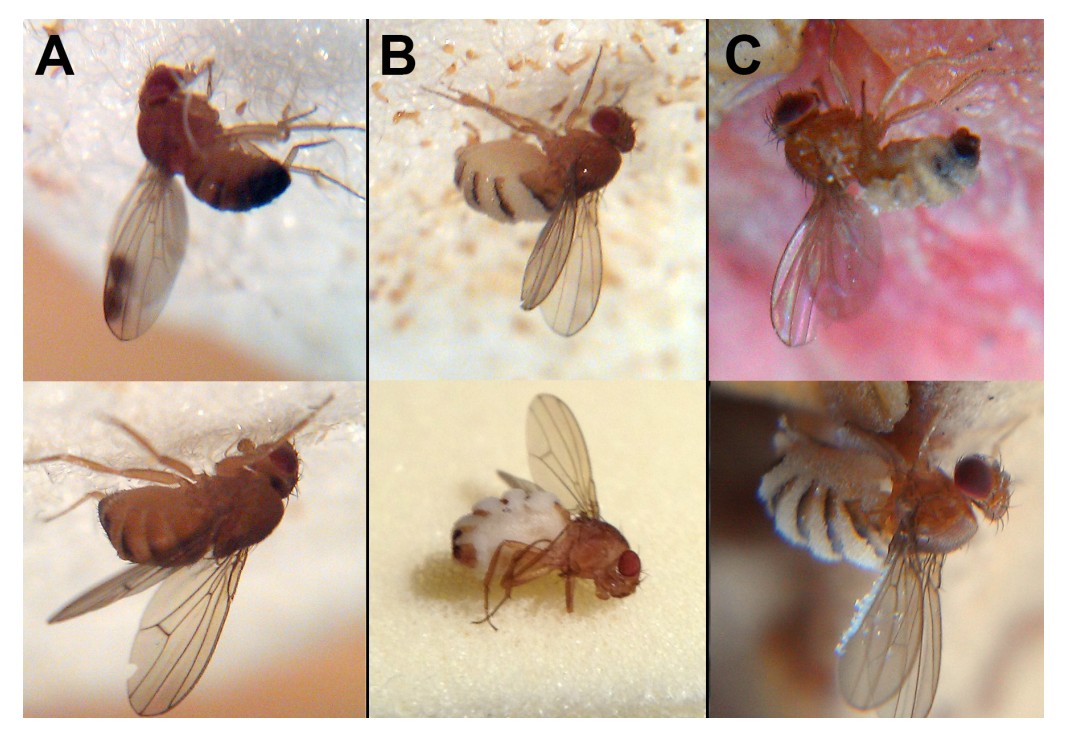

**Figure 1.** Wild drosophilids killed by *Entomophthora muscae* 'Berkeley'. (**A**) Cadavers found among sampled flies 65 min (above) and 40 min (below) after sunset. *E. muscae* 'Berkeley' has not grown through the host cuticle. (**B**) Cadavers found among sampled flies 120 min (above) and 160 min (below) after sunset. *E. muscae* 'Berkeley' has grown through the host cuticle and will soon start to eject conidia. (**C**) Cadavers as discovered in situ in fendel at least 12 hr after sunset. *E. muscae* 'Berkeley' has grown through the host cuticle and ejected conidia, some of which have landed on the cadavers' wings. With the exception of the lower image in panel B, all flies were photographed adhered to the substrate in situ.

DOI: https://doi.org/10.7554/eLife.34414.003

The following figure supplements are available for figure 1:

**Figure supplement 1.** Discovery of *E. muscae* 'Berkeley'.
DOI: https://doi.org/10.7554/eLife.34414.004

**Figure supplement 2.** Confirmation that *E. muscae* 'Berkeley' is growing in vitro.
DOI: https://doi.org/10.7554/eLife.34414.005

**Figure supplement 3.** Time between capture of wild *Drosophila* and death by *E. muscae* infection when housed on Koshland diet.
DOI: https://doi.org/10.7554/eLife.34414.006

counterparts, both morphologically and behaviorally, until they begin to exhibit end-of-life behaviors (*Figure 2A*). Exposed flies bear melanized scars that form following spore entry through the cuticle, which are most apparent when the point of entry is the pale ventral abdomen. However, not all flies that are penetrated by the fungus are successfully infected and killed, as we have observed animals with scarring that survive beyond seven days after exposure, and have found that housing exposed flies on diet with anti- fungal significantly improves survival (*Figure 2—figure supplement 1E*). At 72 hr after exposure and beyond, infected flies generally have more opaque abdomens than uninfected flies due to abundant fungal growth. Under our conditions, ~80% of CantonS flies are killed four to seven days (96–168 hr) after exposure to *E. muscae* 'Berkeley', with the majority of deaths occurring at 96 and 120 hr (*Figure 2B*). While by eye infected animals behave normally until the onset of end-of-life behaviors, analysis of infected fly activity revealed that infected flies exhibit a marked decrease in total activity compared to healthy counterparts beginning about 36 hr before time of death, which presently is the best indication of imminent mortality for a given fly (*Figure 2C*).

On their last day of life, *E. muscae* 'Berkeley'-infected flies make their final movement between 0 and 5 hr before sunset (*Figure 2D*). Taking time of last movement as a proxy for time of death, this observation agrees with reports of *E. muscae* infections in house flies (*Krasnoff et al., 1995*). Also

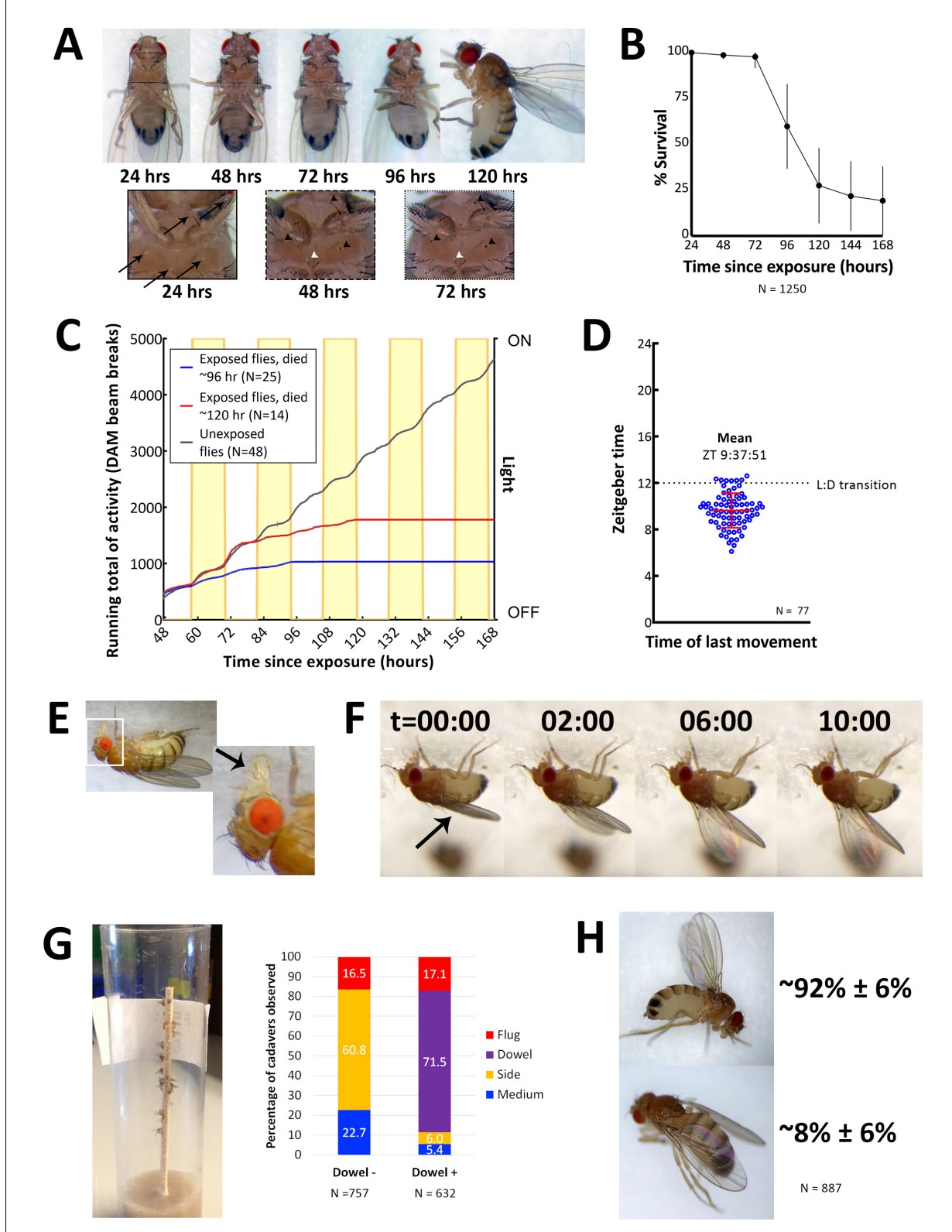

**Figure 2.** Characteristics of *E. muscae* 'Berkeley' infected CantonS. (**A**) A typical female fly over the course of infection. Arrows denote conidia that have landed on the cuticle but not yet bored into the hemolymph. Arrowheads indicate melanization of the fly cuticle that has occurred in response to conidia boring into hemolymph. (**B**) Survival curve for flies infected as per standardized protocol (*Figure 2—figure supplement 1*). Error bars show standard deviation. (**C**) Activity profile of control flies or *E. muscae* 'Berkeley'-infected flies measured using the *Drosophila* activity monitor (DAM).
*Figure 2 continued on next page*

*Figure 2 continued*
Yellow rectangles indicate photophase in the 12:12 light:dark cycle. Error bars show standard deviation. (D) Time of last movement as measured using the DAM. Each blue circle represents the time of last movement observed for one cadaver. Flies were exposed to *E. muscae* 'Berkeley' as per standardized protocol. Dotted line indicates the light-dark transition (L:D transition). Error bars show standard deviation. (E) *E. muscae* 'Berkeley'-infected fly exhibiting proboscis extension tens of minutes before death. Arrow indicates extended proboscis adhered to the surface. Real time footage of an *E. muscae* 'Berkeley'-infected fly undergoing proboscis extension is available as *Video 1*. (F) *E. muscae* 'Berkeley'-infected fly exhibiting wing raising immediately prior to death. Arrow indicates original positioning of wings. Time elapsed is given in minutes:seconds. Real time footage of an *E. muscae* 'Berkeley'-infected fly undergoing wing raising is available as *Videos 2* and *3*. (G) *E. muscae* 'Berkeley'-killed CantonS summited and adhered to a wooden dowel. Graph to the right indicates position of death for flies housed in vials without (Dowel -) or with (Dowel +) a wooden dowel. Percentages of cadavers observed in each location are superimposed on stacked bar plot. (H) Most commonly observed wing positions of *E. muscae* 'Berkeley'-killed CantonS. Complete wing raising is observed in most cadavers; wing lowering is consistently observed in a small fraction of cadavers.
DOI: https://doi.org/10.7554/eLife.34414.007
The following figure supplements are available for figure 2:

**Figure supplement 1.** Optimization of in vivo *E. muscae* infection of CantonS WF *D. melanogaster* under laboratory conditions.
DOI: https://doi.org/10.7554/eLife.34414.008
**Figure supplement 2.** *E. muscae* 'Berkeley'-infected CantonS flies housed in constant darkness do not consistently die in a gated fashion.
DOI: https://doi.org/10.7554/eLife.34414.009

consistent with previous reports, flies exposed to *E. muscae* 'Berkeley' and housed under complete darkness die sporadically throughout the day rather than in a gated fashion (*Figure 2—figure supplement 2*, [*Krasnoff et al., 1995*]). As healthy flies housed for 168 hr in complete darkness maintain circadian rhythm, this suggests that environmental cues to either the fly and/or fungus are required to coordinate the timing of death, as has been previously suggested (*Krasnoff et al., 1995*).

On the last day of life, flies infected with *E. muscae* 'Berkeley' show a precipitous decline. The first portent of imminent death is that they cease to fly. Though they can still walk and are responsive to perturbations (i.e. poking with a paintbrush or jostling their container), they will not take flight. After they have lost the ability or inclination to fly, moribund flies will generally climb upward (if located on a vertical surface) or move towards a vertical surface. Most flies still appear to walk and climb normally, though a small percentage begin to exhibit a shaky and slowed gait as they climb. When provided a thin, wooden dowel as a summiting substrate, approximately 17% more flies die in elevated positions with most flies dying on the dowel itself (*Figure 2G*). Many flies reach elevated positions before they lose the ability (or desire) to continue moving (even when perturbed by the experimenter), but some succumb to immobility before they are able to leave the ground (~95 and ~5%, respectively, when provided with a wooden dowel; ~77 and 23%, respectively, in the absence of a dowel). Interestingly, we have noticed that when drips of medium are present on the side of a vial, flies that die on the side of the vial are preferentially found on these drips. It is unclear if this indicates a preference for the medium as a climbing substrate (versus the smooth plastic of a fly vial), are more easily adhered to than the surrounding plastic or if the flies are attempting to eat until their very last.

Once the moribund flies stops walking, virtually all extend their proboscides until they make contact with the surface on which they are standing (*Figure 2E*). The extension is shaky and occurs slowly relative to extension in response to a nutritive stimulus, and we have observed in multiple instances that the labella of infected flies do not spread as is typically observed when uninfected flies eat (see *Video 1*). As the proboscis is extended, small drops of clear liquid (presumed to be fungal secretions, [*Balazy, 1984*]) can be often observed on the proboscis tip. Typically, once the proboscis has made contact with the surface, many flies move their legs in what appears to be an apparent

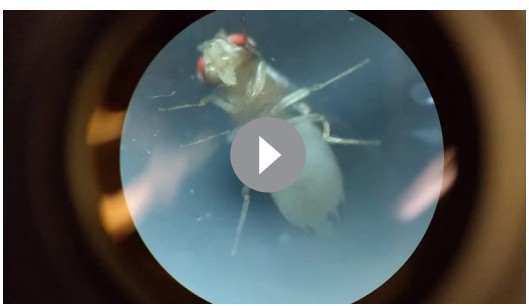

**Video 1.** *E. muscae* 'Berkeley'-infected CantonS fly undergoing end-of-life proboscis extension. Video recorded through the eyepiece of a dissecting microscope on a Nexus 5x phone (Google). Capture and playback are in real time.
DOI: https://doi.org/10.7554/eLife.34414.010

attempt to escape, but the material that emanates from the proboscis is sufficient to keep it anchored in place. After their proboscides have adhered, most (~92%) of flies then begin to raise their wings up and away from their dorsal abdomen (*Figure 2F*). This process takes on the order of 10 min, with wing raising occurring in small bursts, reminiscent of the inflation of a balloon (see *Videos 2* and *3*). Curiously, a persistent minority (~8%) of infected flies die with their wings lowered down onto their abdomen rather than with wings elevated (*Figure 2H*). Applying pressure to the thoraxes of flies like this causes their wings to 'toggle' into the upright position, suggesting that the same muscles may be involved in raising and lowering. Many flies continue to twitch their legs and antennae for several minutes after their wings have reached their final position but will shortly cease moving.

After death, we observe conidiophores, conidia-launching structures, emerge through weak points in the fly's cuticle, as has been described for *E. muscae* infections of other hosts (*Brobyn and Wilding, 1983*). We tracked this process in our flies (*Video 4*). Over the course of several minutes, each conidiophore forms a single primary conidium which initially emerges from the tip of the conidiophore as a small protrusion. The conidium enlarges as material is visibly transferred from the conidiophore into the growing conidium until it is forcibly ejected into the environment.

Using time lapse imaging, we observe that conidia begin to be ejected starting approximately five hours after sunset (*Figure 3A* ii). They continue to fire for several hours at ambient temperature and humidity (*Figure 3A* iii and iv). Conidia form and launch asynchronously within a given cadaver, and some conidiophores form conidia that are never launched.

Using high-speed videography, we captured the motion of conidial ejection (*Figure 3C*), and determined that conidia leave the conidiophore at an initial velocity of ~20 miles per hour (~10 meters/second). These speeds are comparable to those observed in coprophilous fungi, which are among the fastest observed velocities of organisms relative to their size known in the natural world (*Yafetto et al., 2008*). In addition, we obtained high-speed footage of primary conidia landing (*Figure 3D*), which shows that conidia and halo land concurrently, an observation that supports the fungal canon mechanism of spore discharge (*Humber, 1981*). As previously described, a smaller conidium, called a secondary conidium, was observed to form from each of several primary conidia that had landed on non-productive surfaces (host wing or agar substrate) (*Video 8*).

To compare *E. muscae* 'Berkeley' with other reported isolates, we collected primary conidia and measured their key morphological traits (e.g. *Figure 3E*). Our measurements are most similar to primary conidia from *E. muscae sensu strictu* rather than other members of the *E. muscae* species complex (*Supplementary file 2*). *E. muscae sensu strictu* isolates have often been observed to infect house flies (*Musca domestica*) (*MacLeod et al., 1976*; *Keller, 2007*). To gauge if *E. muscae* 'Berkeley' is capable of causing patent infection in hosts outside of Drosophilidae, we exposed house flies to *E. muscae* 'Berkeley'-infected *D. melanogaster* using a modified version of our propagation protocol. We observed flies that died and subsequently sporulated, indicating that they had been killed by the fungus (*Figure 3—figure supplement 1*).

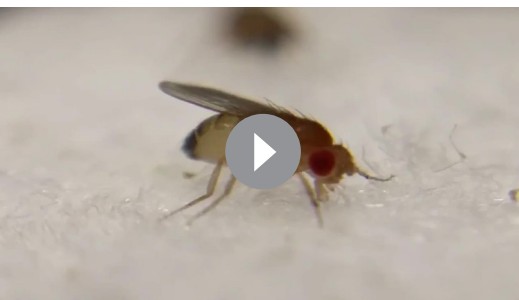 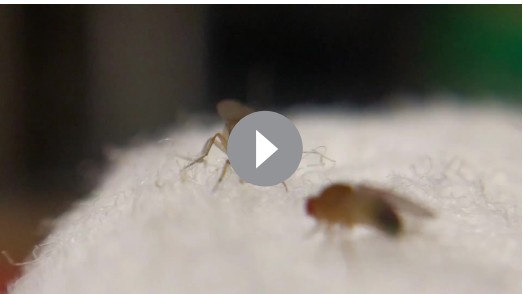

**Video 2.** *E. muscae* 'Berkeley'-infected CantonS fly undergoing end-of-life wing raising, viewed from the side. Video was captured with a Nexus 5x phone (Google) and macro lens (Luxsure). Capture and playback are in real time.
DOI: https://doi.org/10.7554/eLife.34414.011

**Video 3.** *E. muscae* 'Berkeley'-infected CantonS fly undergoing end-of-life wing raising, viewed head-on. Video was captured with a Nexus 5x phone (Google) and macro lens (Luxsure). Capture and playback are in real time.
DOI: https://doi.org/10.7554/eLife.34414.012

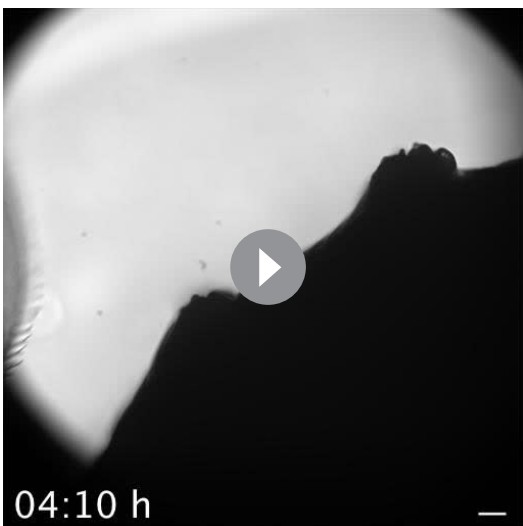

**Video 4.** Formation of primary conidia atop conidiophores. The intersegmental membranes of a fresh *E. muscae* 'Berkeley'-killed cadaver were imaged at 20x magnification every minute for 222 min beginning 4 hr and 10 min after sunset. The time lapse is played back at 10 fps. Scale bar is 50 μm.
DOI: https://doi.org/10.7554/eLife.34414.015

## Transcriptional profiles of *E. muscae* 'Berkeley' and *D. melanogaster* over the course of infection

To gain a first comprehensive look into how *E. muscae* 'Berkeley' infection progresses in *D. melanogaster* at the molecular level, we next measured how transcription changes in both host and fungus at 24 hr time points over 5 days. We knew that in any given exposure vial there are a mix of infected and uninfected animals and faced the complication that, early on, infected flies are phenotypically indistinguishable from uninfected animals. However, we felt confident that we would be able to distinguish infected from uninfected exposed based on the presence of *E. muscae* 'Berkeley' reads in the sample.

We collected six *E. muscae* 'Berkeley' exposed CantonS females at each 24, 48 and 72 hr, three exposed flies at 96 hr, and six fresh cadavers, three at each 96 and 120 hr. In parallel, we collected three CantonS females at each time point from a 'mock' exposure (housed under identical conditions but in the absence of cadavers). We prepared and sequenced mRNA libraries from each individual sampled.

We sought to estimate both the fraction of reads from *D. melanogaster* and *E. muscae* in each sample, as a proxy for the presence of an infection and its progression, and relative levels of *D. melanogaster* and *E. muscae* transcripts in each sample. This process was complicated by the lack of a high quality annotated *E. muscae* genome, and by the presence of various other organisms (yeast in the food, bacteria in the fly- associated microbiome and viruses, as well as common read contaminants especially DNA from the coliphage phiX174 used to calibrate Illumina sequencers).

To overcome these limitations and both get a full accounting of reads (in each sample and to make conservative estimates of *E. muscae* abundance, we implemented a sequential alignment strategy in which we first aligned to *D. melanogaster* (protein-coding mRNAs, then non-coding RNAs, then the genome), then to common contaminants (the phiX174 genome, the genomes of *S. cerevisiae* and the genomes of other fungi, bacteria and viruses found in the sample; see Materials and methods) and then *E. muscae* (a de novo in vivo *E. muscae* transcriptome, a draft *E. muscae* genome annotation and the draft *E. muscae* genome). Only reads that failed to align to the previous references were used as inputs to the next reference, so that, in principle, we only align *E. muscae* reads to *E. muscae*. This strategy was effective in accounting for all reads as fewer than two percent of reads fail to align to any of the references.

We used two different sources of *E. muscae* transcripts - a de novo transcriptome assembled from infected flies and a preliminary annotation of the draft *E. muscae* genome - to ensure that we had as good coverage of the *E. muscae* transcriptome as possible. Because we aligned first to the *E. muscae* transcriptome, there should be limited redundancy between these sets, as only genome annotated transcripts not present in our transcriptome should have a significant numbers of reads aligned to them. The value of this process is evinced by the fact that a significant number of reads aligned to the annotated transcripts after alignment to the de novo transcriptome (approximately 10 percent as many as aligned to the transcriptome). The number of reads aligning to the *E. muscae* transcriptome and annotated transcripts are highly correlated across samples, suggesting that they are giving complementary information about the same source of reads. Subsequent analyses presented here use only reads that aligned to the annotated *D. melanogaster* mRNAs as representing *D. melanogaster* and the combination of reads aligning to our *E. muscae* 'Berkeley' transcriptome and annotated transcripts from the *E. muscae* genome to represent *E. muscae*.

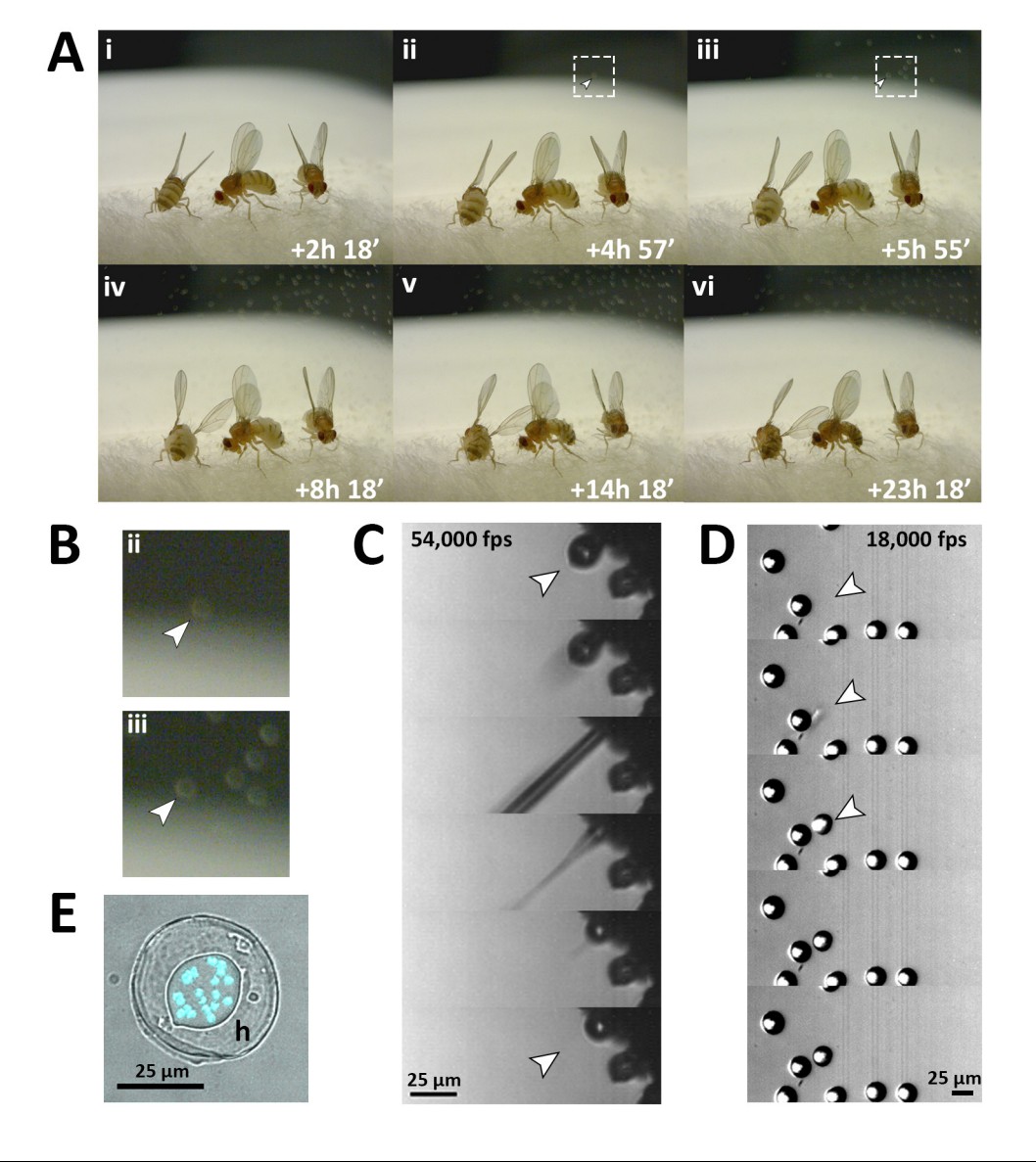

**Figure 3.** Fungal transmission from *E. muscae* 'Berkeley' killed cadavers. (**A**) Sporulation time lapse in. (**E**) muscae 'Berkeley' killed cadavers. Time listed in each frame is the time that has elapsed since the light- dark transition. One image was taken every minute for ~24 hr with three cadavers situated on a cotton plug at ambient temperature and humidity. The arrowhead in frames ii, iii and their respect enlarged insets in B indicates the first primary conidium observed to land on the camera's lens, indicating the start of conidial ejection (i.e. sporulation). Animated time lapse available as *Video 5*. (**B**) Enlarged regions outlined in A. (**C**) Time lapse of the ejection of a primary conidium from a sporulating cadaver as captured at 54,000 frames per second (fps). Arrowheads indicate conidium that launches and the vacant conidiophore that remains after launch. Animated time lapse available as *Video 6*. (**D**) Time lapse of a primary conidium landing on the lid of a coverslip as captured at 18,000 fps. The conidium lands as one complete unit, supporting the fungal cannon mechanism of primary conidium ejection in *E. muscae*. Arrowheads indicates the position where the primary conidium lands (the conidium simply appears from one frame to the next). Animated time lapse available as *Video 7*. (**E**) Primary conidium adhered to glass coverslip and stained with fluorescent nuclear dye (Hoechst 33342). The conidium is surrounded by a halo of co-ejected material (h).

DOI: https://doi.org/10.7554/eLife.34414.013

The following figure supplement is available for figure 3:

**Figure supplement 1.** House flies (*M. domestica*) infected with *E. muscae* 'Berkeley' five to eight days after exposure to *E. muscae* 'Berkeley'-killed *D. melanogaster* cadavers.

*Figure 3 continued on next page*

*Figure 3 continued*

DOI: https://doi.org/10.7554/eLife.34414.014

We first examined the proportion of reads that aligned to host or fungus in each of our time course samples taken only the reads aligning to these references into consideration (*Figure 4A*). We observed that *E. muscae* 'Berkeley' reads are generally not detected until 72 hr after exposure at which point a significant portion of the total reads align to the *E. muscae* 'Berkeley' references. This likely reflects that the fungus is at low titer until between 48 and 72 hr, consistent with our observation that *E. muscae* 'Berkeley' rRNA is not reliably detectable by endpoint reverse transcription PCR until at least 72 hr after exposure (*Figure 4—figure supplement 1*). Cadavers vary in the proportion of *E. muscae* mRNA compared to *D. melanogaster* mRNA, which likely negatively correlates with the amount of remaining, intact fly tissue. Strikingly, two of our cadavers show only trace amounts of *D. melanogaster* RNA at the point of sampling.

We next surveyed gene expression in host and fungus across all samples. As different exposed individuals collected at the same nominal time point may bear different loads of *E. muscae* 'Berkeley', we reasoned that it would be most informative to order our samples based on *E. muscae* 'Berkeley' titer, which we approximated using the proportion of reads that aligned to *E. muscae* 'Berkeley' of total reads aligned to either the *E. muscae* 'Berkeley' or *D. melanogaster* references (*Figure 4B–D*). Using our de novo *E. muscae* 'Berkeley' transcriptome as a reference, we observed the bulk of transcripts were not expressed until three days after exposure (Group i), which is likely a consequence of the fungus being low abundance until this time point (*Figure 4B*).

The predicted translated products of 42% (1545 out of 3678) of the transcripts in Group i show homology to protein entries in the Uniprot database with an e-value less than 1e-40. Based on these matches, the putative functions of Group i translated products include a variety of processes common to eukaryotic cells including transcription, translation, ATP production, chromatin remodeling, cytoskeleton formation, signaling (kinases, phosphatases, G-proteins), ionic, molecular and vesicular transport and secretion, and targeted degradation of proteins and mRNA (*Supplementary file 3*). In addition, Group i includes a transcript whose translated product has homology to white collar protein, a photoreceptor and transcriptional regulator of the molecular circadian clock gene *frq* in *Neurospora crassa* (*Lee et al., 2003*; *Ballario et al., 1996*), as well as a transcript whose translated product is homologous to the light-sensitive gene cryptochrome (*Lin and Todo, 2005*). This suggests that *E. muscae* 'Berkeley' may also have the ability to sense light and maintain a molecular clock. Group i also includes two transcripts whose translated products have homology to argonaute, a key protein in the RNA-induced silencing complex (RISC) (*Fagard et al., 2000*), as well as transcripts whose products are homologous to aspartic proteases, which are required for the mobilization of retrotransposons. The latter is consistent with our observation that the *E. muscae* 'Berkeley' genome (NCBI PRJNA479887) is at least 85% repeat content, most of that consisting of retrotransposons.

Interestingly, there are two groupings of genes (Groups ii and iii) that demonstrated patterns that cannot be explained merely by fungal abundance in the samples. Group ii consists of genes that are expressed during the later phases of growth in the living host but are turned off after the fly is killed; 26.6% (135 out of 508) of these translated transcripts show homology to

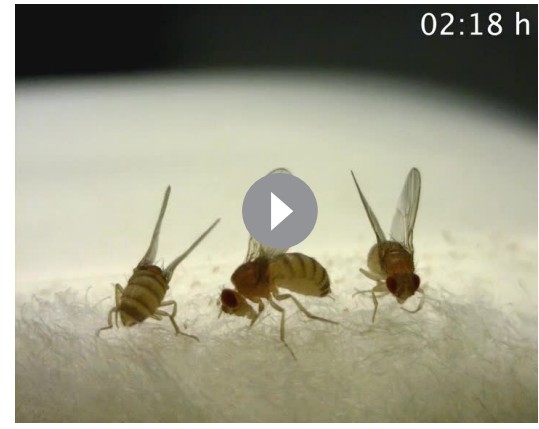

**Video 5.** Animated time lapse of *E. muscae* 'Berkeley'-infected CantonS cadavers undergoing spore production and ejection. Sporulation time lapse in *E. muscae* 'Berkeley'-killed cadavers. Time listed at the top right corner of each frame is the time that has elapsed since the light-dark transition. One image was taken every minute for ~24 hr with three cadavers situated on a cotton plug at ambient temperature and humidity. Images are played back at 10 fps.
DOI: https://doi.org/10.7554/eLife.34414.016

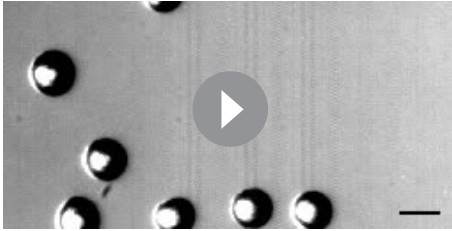

**Video 6.** A primary conidium is ejected from a conidiophore formed in an *E. muscae* 'Berkeley'- killed cadaver. Arrowheads indicate conidium that launches and the vacant conidiophore that remains after launch. Video was captured at 54,000 frames per second (fps) at 5x magnification; frames are played back at 5 fps. Scale bar is 25 μm.
DOI: https://doi.org/10.7554/eLife.34414.017

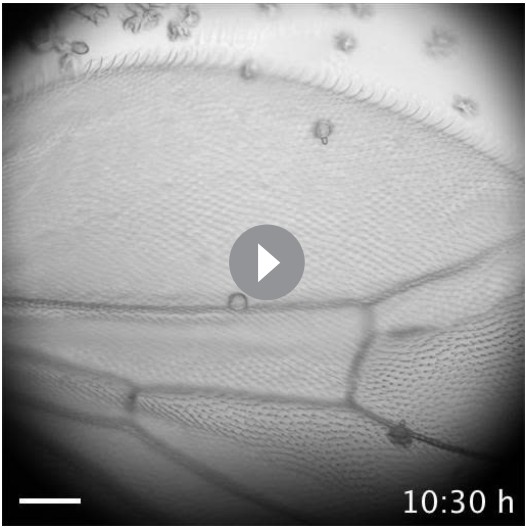

**Video 7.** A primary conidium lands on the lid of a polystryene petri dish. The conidium lands as one complete unit, supporting the fungal cannon mechanism of primary conidium ejection in *E. muscae*. Video was captured at 18,000 fps at 5x magnification; frames are played back at 5 fps. Scale bar is 25 μm.
DOI: https://doi.org/10.7554/eLife.34414.018

known proteins at e-values less than 1e-40. Notable homology in Group ii includes a transcript that encodes a homolog of trehalase, which degrades the sugar trehalose, the most abundant sugar in insect hemolymph (*Bedford, 1977*), a transcript encoding patatin, an enzyme that which degrades lipids (*Senda et al., 1996*), and a transcript that encodes a homolog to aquaporin, a channel that transports water across cell membranes (*Agre et al., 1993*). Multiple transcripts in Group ii encode homologs of chitin synthases, which is consistent with *E. muscae* 'Berkeley's transition from protoplastic to cell-walled growth before departing the host, as well as a homolog of Dibasic-processing endoprotease, which is involved in the maturation of peptide pheromones. Interestingly, Group ii also includes a transcript whose translated product is homologous to the voltage-dependent calcium channel Cch1 in *Saccharomyces cerevisiae*, which is known to respond to environmental stress as well as mating hormones (*Fischer et al., 1997*; *Peiter et al., 2005*). This is noteworthy given recent evidence that recombination occurs in *E. muscae* but is not yet understood when or how frequently *E. muscae* exchanges genetic material (*Gryganskyi et al., 2013*).

Group iii consists of genes that are not expressed until after the fungus has killed the host. Only 10.8% (44 out of 407) of these translated transcripts show homology to known proteins with an e-value less than 1e-40. Among these, notable annotations include metalloregulation (metal tolerance, metalloproteases) as well as proteases, including a homolog to kexin, a protease that processes prohormones and could possibly be involved in fungal mating or communication (*Fuller et al., 1989*). Additionally, one transcript encodes a homolog of insulin-degrading enzyme, which could play a role in utilizing the remaining resources in the deceased host. We next aligned reads that did not align to our *E. muscae* 'Berkeley' transcriptome to annotated transcripts in our *E. muscae* 'Berkeley' genomic reference and estimated transcript abundance (*Figure 4C*). The resultant expression pattern is similar to that of many transcripts from our transcriptome not included in Groups i–iii (*Figure 4B*), and generally reflects a weak inverse correlation between inferred fungal titer and expression.

We examined host gene expression patterns across all of our samples, again ordering samples based on the proportion of *E. muscae* 'Berkeley' aligned reads among all total and clustering genes by expression pattern (*Figure 4D*, *Supplementary file 4*). Group i contains genes

**Video 8.** Formation of secondary conidia from off-target primaries. The wing of an *E. muscae* 'Berkeley'-killed cadaver was imaged at 20x magnification every minute for 361 min beginning 10 hr and 30 min after the light-dark transition. The time lapse is played back at 10 fps. Scale bar is 100 μm.
DOI: https://doi.org/10.7554/eLife.34414.019

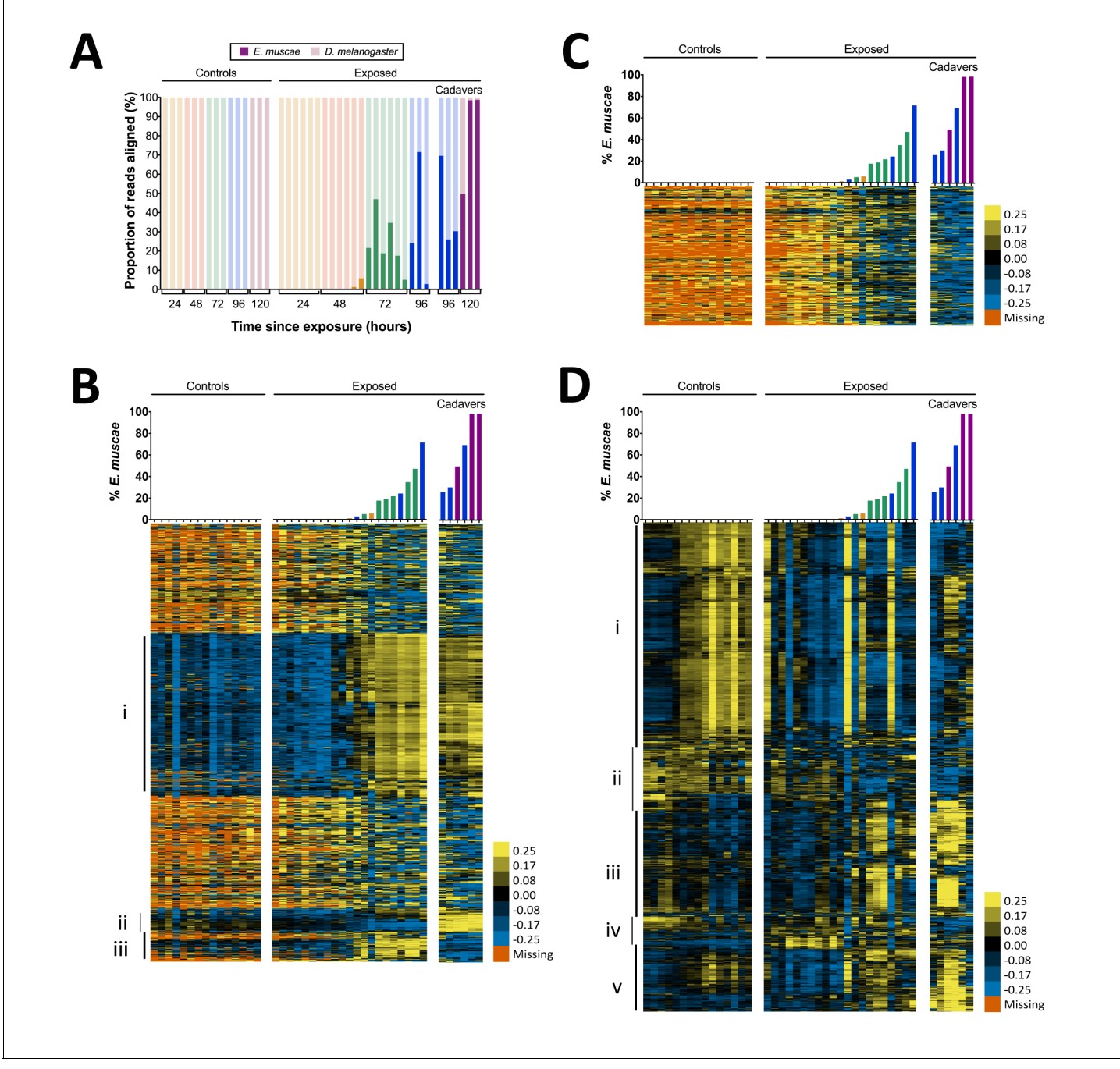

**Figure 4.** Gene expression time course of *E. muscae* 'Berkeley'-infected CantonS flies. (A) Proportion of reads aligned to *D. melanogaster* reference (mRNA) versus *E. muscae* 'Berkeley' references (de novo transcriptome and de novo annotated mRNAs in genome) using bowtie2. Samples are separated into controls (healthy animals who were mock exposed), exposed (animals who exposed to *E. muscae* 'Berkeley' and were alive at the time of sampling) and cadavers (animals who were killed by *E. muscae* 'Berkeley', dead at the time of sampling) and are color-coded according to the time point at which they were collected (i.e. 24, 48, 72, 96 or 120 hr). (B) *E. muscae* 'Berkeley' expression data from *E. muscae* 'Berkeley'- exposed and cadaver samples calculated from alignment to de novo *E. muscae* transcriptome. Complete linkage hierarchical clustering by gene was performed in Gene Cluster 3.0 after filtering out genes that are not expressed at least at ten TPM in at least three out of 42 samples (9500 transcripts total), then log transforming and centering on the mean value for each transcript. Samples are ordered by percentage of *E. muscae* 'Berkeley' reads as a fraction of the total reads aligned (above). The scale bar for the heatmap is given to the right of the plot. (C) *E. muscae* 'Berkeley' expression data from *E. muscae* 'Berkeley'-exposed and cadaver samples calculated from alignment to de novo *E. muscae* genome. Complete linkage hierarchical clustering by gene was performed in Gene Cluster 3.0 after filtering out genes that are not expressed at least at two TPM in at least three out of 42 samples (3104 transcripts total), then log transforming and centering on the mean value for each transcript. Samples are ordered by percentage of *E. muscae*

*Figure 4 continued on next page*

*Figure 4 continued*

'Berkeley' reads as a fraction of the total reads aligned (above). The scale bar for the heatmap is given to the right of the plot. (D) *D. melanogaster* expression data from control, *E. muscae* 'Berkeley'- exposed and *E. muscae* 'Berkeley'-killed cadavers. Complete linkage hierarchical clustering by gene was performed in Gene Cluster 3.0 after filtering out genes that are not expressed at least at two TPM in at least three out of 42 samples (10,746 transcripts total), then log transforming and centering on the mean value for each transcript. Samples are ordered by percentage of *E. muscae* 'Berkeley' reads as a fraction of the total reads aligned (above). The scale bar for the heatmap is given to the right of the plot. Heatmaps are scaled identically between panels (B-D).

DOI: https://doi.org/10.7554/eLife.34414.020

The following figure supplements are available for figure 4:

**Figure supplement 1.** Reverse-transcription PCR of *E. muscae* 'Berkeley' ITS sequence in exposed and control flies.

DOI: https://doi.org/10.7554/eLife.34414.021

**Figure supplement 2.** Genes exhibiting differential expression between flies after mock exposure or exposure to *E. muscae* between living flies collected at individual time points.

DOI: https://doi.org/10.7554/eLife.34414.022

**Figure supplement 3.** Expression of immune genes over the course of infection of *D. melanogaster* by *E. muscae* 'Berkeley'.

DOI: https://doi.org/10.7554/eLife.34414.023

that are weakly expressed over the bulk of *E. muscae* 'Berkeley' infection and after death, but are highly expressed in control samples after 48 hr. Group i is enriched for genes involved in general metabolism (translation, transcription and DNA regulation) as well as oogenesis. Group ii contain genes that are expressed in the first 48 hr after exposure to *E. muscae* 'Berkeley' and that are generally expressed in all controls. These genes play roles predominantly in metabolism of polysaccharides, fatty acids, hormones pigments. Group iii contains genes that are expressed in early control samples (24–28 hr) and in later stages of infection and death. Interestingly, many of these genes are involved in sensory pathways (including hearing, vision, circadian rhythm) and basic neuronal processes (e.g. synaptic development and transmission, ion transport). Group iv contains genes that are expressed only in early controls, or not expressed in controls at all, but are expressed strongly in early exposed samples with lingering expression throughout late infection and death. These genes are most highly enriched for immune functions and energy extraction from carbohydrates (citric acid cycle and electron transport chain). Group v consists of genes that are expressed at later time points in both control and exposed flies, enriched for a diverse set of genes involved in stress response, cell signaling, chemical stimuli and morphogenesis.

Following our initial overview of host transcription, we next looked at genes that were consistently different between control and exposed flies between 24–96 hr (*Figure 5*, *Supplementary file 4*). We excluded cadaver samples from these analyses because the animals are dead, and variations in gene expression could be confounded by mRNA degradation. One-way ANOVA analysis between exposed and control animals from 24 to 96 hr demonstrated that genes that are under-expressed in exposed animals are enriched for a variety of metabolic processes, including catabolism of oligosaccharides, purine nucleobases and amino acids, and specifically, metabolism of glutamine. Interestingly, glutamine is synthesized from the Kreb's cycle intermediate alpha-ketoglutarate. In times of starvation, the cell would be expected to prioritize generating ATP via the Kreb's cycle over glutamine synthesis. The idea that the fly is starving is consistent with these enrichments and also with the observation that basic cell metabolism (macromolecule synthesis) is substantially decreased at 72 hr (*Figure 4—figure supplement 2*, *Supplementary file 5*).

The same analysis shows that genes that are over-expressed in exposed animals are enriched for immune function, including the melanization defense response and Toll-dependent pathways. *E. muscae* infection relies on boring through the host cuticle (*Brobyn and Wilding, 1983*) which should elicit an initial melanization response, consistent with our observation. However, it is generally thought that *E. muscae* evades the host immune response once inside the fly because it grows without a cell wall (i.e. protoplastically) and therefore does not present antigens that can alert the fly immune system to infection (*Boomsma et al., 2014*; *De Fine Licht et al., 2017*). Examining expression patterns of all genes annotated as having immune function, we see a large induction of immune gene expression at 24–48 hr which includes genes both involved in the melanization response and genes that specifically respond to fungal infection. In addition, we see overexpression of several groups of immune genes compared to uninfected controls that persists into late infection (72 and 96

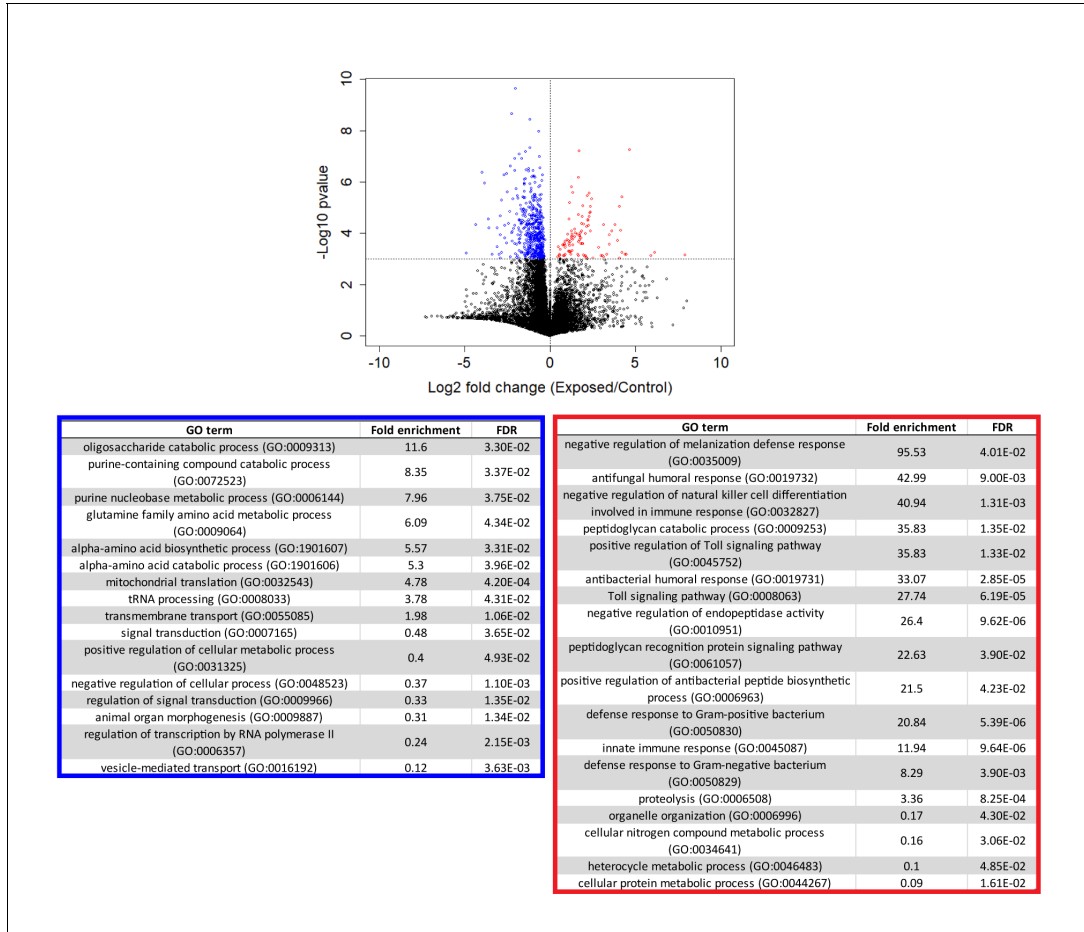

| GO term | Fold enrichment | FDR |
|---|---|---|
| oligosaccharide catabolic process (GO:0009313) | 11.6 | 3.30E-02 |
| purine-containing compound catabolic process (GO:0072523) | 8.35 | 3.37E-02 |
| purine nucleobase metabolic process (GO:0006144) | 7.96 | 3.75E-02 |
| glutamine family amino acid metabolic process (GO:0009064) | 6.09 | 4.34E-02 |
| alpha-amino acid biosynthetic process (GO:1901607) | 5.57 | 3.31E-02 |
| alpha-amino acid catabolic process (GO:1901606) | 5.3 | 3.96E-02 |
| mitochondrial translation (GO:0032543) | 4.78 | 4.20E-04 |
| tRNA processing (GO:0008033) | 3.78 | 4.31E-02 |
| transmembrane transport (GO:0055085) | 1.98 | 1.06E-02 |
| signal transduction (GO:0007165) | 0.48 | 3.65E-02 |
| positive regulation of cellular metabolic process (GO:0031325) | 0.4 | 4.93E-02 |
| negative regulation of cellular process (GO:0048523) | 0.37 | 1.10E-03 |
| regulation of signal transduction (GO:0009966) | 0.33 | 1.35E-02 |
| animal organ morphogenesis (GO:0009887) | 0.31 | 1.34E-02 |
| regulation of transcription by RNA polymerase II (GO:0006357) | 0.24 | 2.15E-03 |
| vesicle-mediated transport (GO:0016192) | 0.12 | 3.63E-03 |

| GO term | Fold enrichment | FDR |
|---|---|---|
| negative regulation of melanization defense response (GO:0035009) | 95.53 | 4.01E-02 |
| antifungal humoral response (GO:0019732) | 42.99 | 9.00E-03 |
| negative regulation of natural killer cell differentiation involved in immune response (GO:0032827) | 40.94 | 1.31E-03 |
| peptidoglycan catabolic process (GO:0009253) | 35.83 | 1.35E-02 |
| positive regulation of Toll signaling pathway (GO:0045752) | 35.83 | 1.33E-02 |
| antibacterial humoral response (GO:0019731) | 33.07 | 2.85E-05 |
| Toll signaling pathway (GO:0008063) | 27.74 | 6.19E-05 |
| negative regulation of endopeptidase activity (GO:0010951) | 26.4 | 9.62E-06 |
| peptidoglycan recognition protein signaling pathway (GO:0061057) | 22.63 | 3.90E-02 |
| positive regulation of antibacterial peptide biosynthetic process (GO:0006963) | 21.5 | 4.23E-02 |
| defense response to Gram-positive bacterium (GO:0050830) | 20.84 | 5.39E-06 |
| innate immune response (GO:0045087) | 11.94 | 9.64E-06 |
| defense response to Gram-negative bacterium (GO:0050829) | 8.29 | 3.90E-03 |
| proteolysis (GO:0006508) | 3.36 | 8.25E-04 |
| organelle organization (GO:0006996) | 0.17 | 4.30E-02 |
| cellular nitrogen compound metabolic process (GO:0034641) | 0.16 | 3.06E-02 |
| heterocycle metabolic process (GO:0046483) | 0.1 | 4.85E-02 |
| cellular protein metabolic process (GO:0044267) | 0.09 | 1.61E-02 |

**Figure 5.** Genes that are consistently over or under-expressed compared to controls over the first 96 hr after exposure to *E. muscae* 'Berkeley'. Top: Volcano plot for all genes over the first 96 hr after exposure. P-value is determined by ANOVA grouping 24–96 hr control vs. 24–96 hr exposed samples. Genes with p-value below 0.001 are shown in color. Bottom: PANTHER Gene Ontology (GO)-term analysis (complete biological process) of genes overexpressed (red) or under-expressed (blue) in exposed animals compared to controls. FDR column lists the p-value from a Fisher's Exact test with false discovery rate correction for multiple testing.
DOI: https://doi.org/10.7554/eLife.34414.024

hr) and even into death (96 and 120 hr) (*Figure 4—figure supplement 3*). These data suggest that the initial immune response may not be strictly limited to wound repair and show that the host immune system remains engaged throughout infection.

### *E. muscae* 'Berkeley' is present in the fly nervous system 48 hr after exposure

To better understand the process of *E. muscae* 'Berkeley' infection in *D. melanogaster*, we next used a histological approach to examine the interior anatomy of exposed flies. Analogous to the transcriptomic time course, we collected adult flies (a mix of 50 males and females) every 24 hr for the first 168 hr after *E. muscae* 'Berkeley' or mock exposure. Flies were fixed before embedding and sectioning in paraffin then stained with Safranin O/Fast Green (SFG), a contrast staining method that facilitates the differentiation of fungal versus host cells (Richard Humber, personal communication), though is more commonly used for plant histology. We identified *E. muscae* 'Berkeley' morphology by examining *E. muscae* 'Berkeley'-killed hosts. While there is slide-to-slide variability in the exact hue for a given tissue stained with SFG, generally, we observed that SFG-stained *E. muscae* 'Berkeley' hyphal bodies have nuclei that stain red (or dark purple) and cytoplasm that stains purple (*Figure 6*). *E. muscae* 'Berkeley' nuclei are consistently sized throughout the host which helps in distinguishing them from host *D. melanogaster* cells.

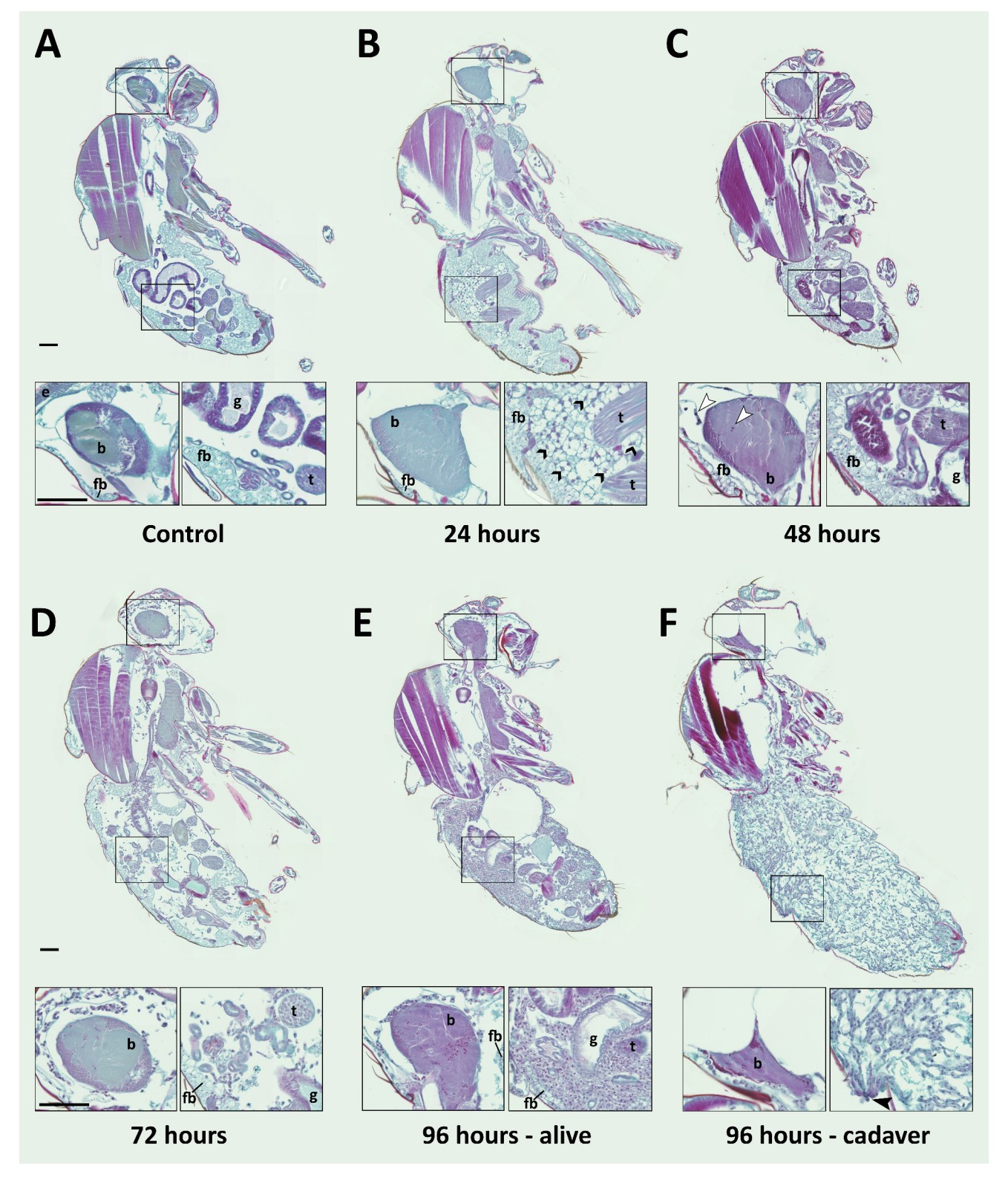

**Figure 6.** *E. muscae* 'Berkeley' is consistently present in the nervous system starting 48 hr after exposure. CantonS flies were exposed or mock-exposed (control) to *E. muscae* 'Berkeley' starting 3–5 hr after the light-dark transition and were subsequently sampled at 24, 48, 72 or 96 hr for histological analysis. For each time point, 4–6 individual, paraffin-embedded flies were sectioned at 8 μm, stained using Safranin O/Fast Green to identify fungal morphology and location and imaged at 20x magnification (Zeiss Axio Scan.Z1 slide scanner). Representative images from each time point are shown. *Figure 6 continued on next page*

*Figure 6 continued*

Only male flies are shown here for ease of comparison. No differences in the progression of the infection were observed between males and females. An inset of the brain and the abdomen are shown for each sample. (A) Uninfected fly with major anatomical features labeled as follows: e - eye, b - brain, g - gut, t - testes, fb - fat body. (B) At 24 hr after exposure there is significant inflammation observed in the abdominal fat body (black arrowheads label a few clear examples); the nervous system is devoid of fungal cells. (C) At 48 hr after exposure *E. muscae* 'Berkeley' cells are present in the brain (white arrowheads) and/or ventral nerve cord (VNC) of all but one sample where *E. muscae* 'Berkeley' cells abut but have not yet entered brain. *E. muscae* 'Berkeley' cells are sparsely observed in the abdominal and/or thoracic hemolymph at 48 hr. The gut and testes are not invaded by fungus. (D) At 72 hr after exposure, *E. muscae* 'Berkeley' can be found throughout the body cavity and the amount of visible fat body has decreased. *E. muscae* 'Berkeley' titers have increased in the nervous system. (E) In a living fly at 96 hr after exposure (the first point at which a fly may be killed by *E. muscae* 'Berkeley' infection), fungus occupies virtually all available volume in the hemolymph. *E. muscae* 'Berkeley' titers have increased in the nervous system, gut and gonads remain uninvaded. (F) In an *E. muscae* 'Berkeley'-killed fly (cadaver) at 96 hr after exposure, only traces of host organs remain in the abdomen and the nervous system has been considerably degraded. No fat body cells are observed. *E. muscae* 'Berkeley' cells differentiate into conidiophores, cell-walled structures that will pierce through weak points of the cuticle to produce and launch infectious conidia. Black scale bars are 100 μm. All living animals shown are males; cadaver's sex is undetermined (the gonads have been consumed by the fungus).

DOI: https://doi.org/10.7554/eLife.34414.025

We then carefully examined SFG-stained sections from exposed and control flies to determine where the fungus resides and how fly tissue is impacted over the course of infection (*Figure 6*). As we observed no difference in fungal localization between males in females, all samples from a given time point are described regardless of sex. In control animals, sagittal sections consistently show abundant fat body cells in the abdomen surrounding the gut and gonads. Fat body is also apparent, though less abundant, in the head and thorax. The thorax is predominantly occupied by muscles (generally staining red), which are also observed in the legs. At 24 hr after exposure, we observed fat body inflammation in the abdomen, with all other tissues indistinguishable from controls. Though fat body inflammation indicates that the immune system is responding to the fungus, we could not unambiguously identify the fungal cells anywhere in the body cavity at this time point. We must therefore conclude that the morphology of fungal cells initially after entry into the hemolymph is distinct from that observed at later time points (48 hr and beyond).

At 48 hr after exposure, fungal cells are consistently observed in the brain and/or ventral nerve cord (VNC; 4 out of 5 samples). In the one case where fungus had not invaded the nervous system, hyphal bodies were apparent immediately adjacent to the brain, abutting the blood brain barrier. A handful of fungal cells are also observed in the abdomen or thorax, with some samples showing fat body inflammation as in 24 hr samples. At 72 hr after exposure, fungal growth is apparent throughout the body cavity and some fat body inflammation can still be observed. The fat body is depleted compared to earlier time points and fungus is apparent between muscle fibers, but the gut and gonads all appear indistinguishable from controls. In addition, fungal titers increase in the brain and VNC. In infected animals that survived 96 hr, fungal growth is rampant throughout the entire body cavity (head, thorax and abdomen), with the fat body substantially depleted and fungus residing between muscle fibers. There is no apparent damage to the gut or gonads. Occasional fat body inflammation can still be observed; fungal titers continue to increase in the brain and VNC. In *E. muscae* 'Berkeley'-killed cadavers collected two to four hours following the light- dark transition, fungus is apparent throughout the body cavity, especially in the abdomen. The gut and gonads have been completely consumed by the fungus, the brain has begun to be degraded and the muscles are largely intact.

To confirm that the morphologies observed in the nervous system at 48 hr after exposure and beyond were *E. muscae* 'Berkeley', we used fluorescence in situ hybridization (FISH) to specifically label *E muscae* Berkeley cells within the context of an infected fly. By performing FISH with a fluorescently-labeled DNA probe targeting the most abundant repeated 18mer in the *E. muscae* 'Berkeley' genome (~11,000 copies), we verified that *E. muscae* 'Berkeley' is present in the brain and VNC in infected animals (*Figure 7*).

Following our observation that *E. muscae* 'Berkeley' invades the nervous system, we wondered if we could identify specific neuronal pathways targeted by the fungus by analyzing gene expression from fly brains during infection. We performed a pilot experiment in which we sequenced RNA from individual dissected brains of animals at 24, 48 and 72 hr following exposure to *E. muscae* 'Berkeley' with paired unexposed controls, using the same alignment pipeline as described for our whole fly

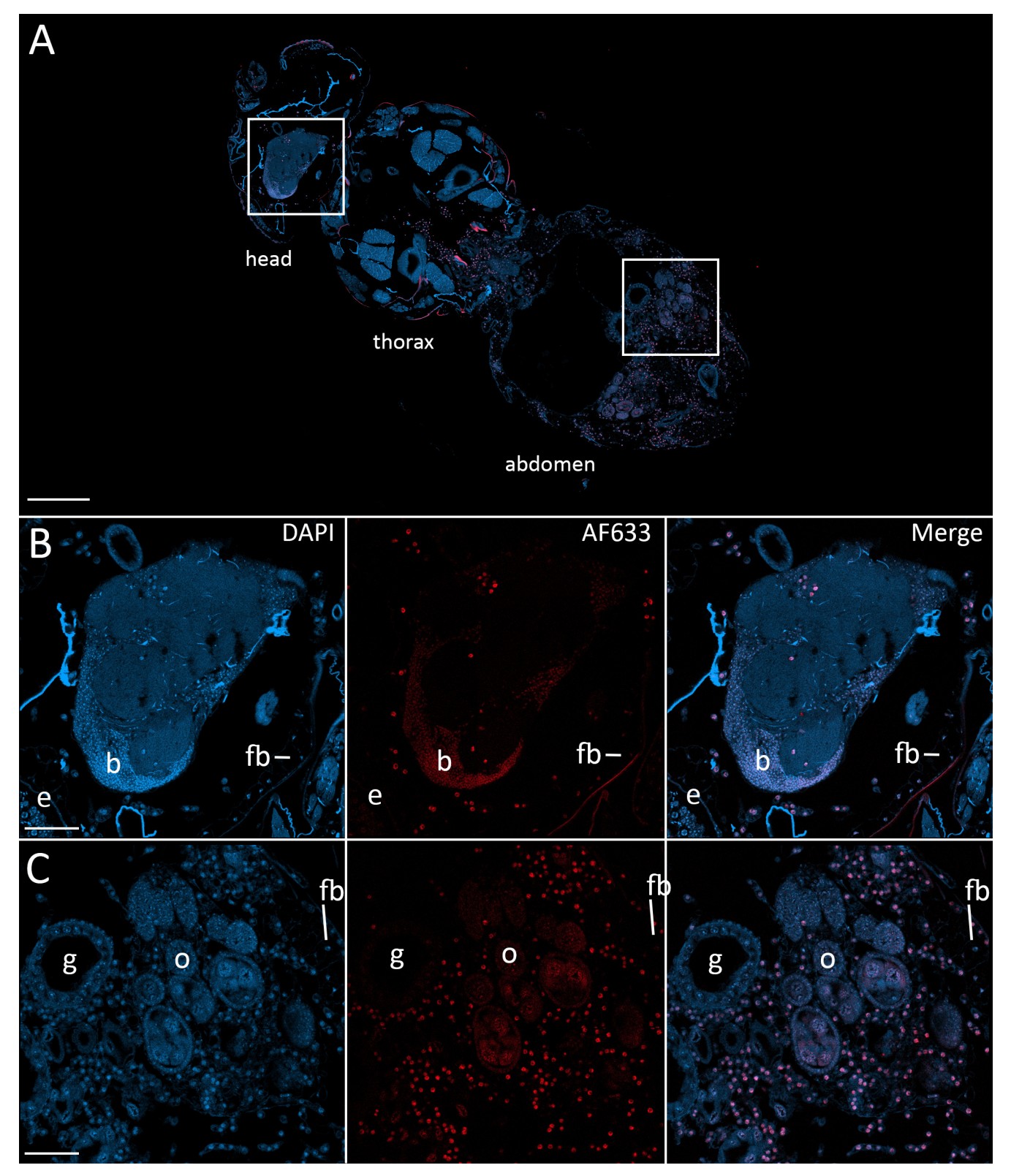

**Figure 7.** Fluorescence in situ hybridization confirms that *E. muscae* 'Berkeley' resides in the nervous system during infection. CantonS flies were exposed or mock-exposed (control) to *E. muscae* 'Berkeley' starting 3–5 hr after the light:dark transition and were subsequently sampled at 24, 48, 72 or 96 hr for histological analysis. For time points 48 hr and later, at least 3 individual, paraffin-embedded flies were sectioned at 8 μm and subjected to

*Figure 7 continued on next page*

*Figure 7 continued*

FISH with an *E. muscae* 'Berkeley'-specific 18mer DNA probe labeled at the 5' end with AlexaFluor633. Sections were imaged at 40x magnification on a confocal fluorescent microscope (Zeiss 800 LSM). (**A**) Pseudo-coronal section of a female sampled 96 hr after infection stained with an *E. muscae* 'Berkeley'-specific probe and DAPI. Regions shown at higher detail in B and C located are denoted by white boxes. Scale bar is 200 µm. (**B**) Enlargement of top region in A showing each DAPI, AlexaFluor633 and merged images of this area. *E. muscae* 'Berkeley' nuclei are strongly labeled and can be observed inside the host neuropil. Fungal nuclei are also observed in the head outside of the brain. Scale bar is 50 µm. Major anatomical features labeled as follows: e - eye, b - brain, fb - fat body. (**C**) Enlargement of bottom region in A showing each DAPI, AlexaFluor633 and merged images of this area. *E. muscae* nuclei are observed in abundance outside of gut and ovaries. Scale bar is 50 µm. Major anatomical features labeled as follows: g - gut, o - ovaries, fb - fat body.

DOI: https://doi.org/10.7554/eLife.34414.026

samples. Three brains were sequenced for each treatment, except for control animals at 48 hr, for which two brains were sequenced. To our surprise, we observed sparse differences in the host transcriptional profiles between control and exposed samples when looking at individual time points (i.e. 24, 48 or 72 hr) or all time points combined (*Figure 8*). There were a small number of *D. melanogaster* genes that showed significant changes in expression between control and exposed flies, but they did not collectively suggest a specific region of the brain or pathway. The upregulation of *fruitless*, which is involved in a number of sex-specific behaviors, and *timeless*, which is involved in the circadian clock, is intriguing, but the small number of changes in host expression suggests that if there are important expression differences in the brain they are likely limited to small numbers of cells and will only be revealed by single-cell expression experiments.

Still, we see clear evidence that *E. muscae* 'Berkeley' is present in the brain and observe an increase in the expression of several fungal transcripts at 72 hr compared to earlier time points (*Figure 8—figure supplement 1*). The predicted translated products of 40.5% (1419 out of 3500) of *E. muscae* 'Berkeley' transcripts that show increased expression have tentative annotations (BLAST hits with e-values less than 1e-40; *Supplementary file 6*). Many of these genes are involved in core eukaryotic cellular processes (transcription, translation, molecular transport, cytoskeletal regulation etc) that often increase in expression in actively dividing cells. However, several of these genes have more intriguing annotations, including an adiponectin receptor (a hormone receptor), neuroendocrine convertase and kexin (both involved in peptide pheromone maturation), phenylalanine-4-hydroxylase (an enzyme necessary for the production of tyrosine which in turn is a building block for hormones, including serotonin), a putative oligopeptide transporter and voltage-gated potassium channels.

While these genes have fungal-specific functions, their presence in the *D. melanogaster* brain provides various means by which the fungus could communicate with neurons in their own chemical language, and highlights the potential of having a system with unparalleled neurogenetic tools with which to investigate it.

## Discussion

A diverse array of microbes has independently evolved the ability to alter animal behavior, and many of these fungi parasitize insects. The prospect of understanding how they do this is intrinsically motivating, and potentially of great practical value as a means to understand how animal behaviors are generated and how we might manipulate them in disease therapy, pest control and other contexts. While researchers since the mid-19th century have continued to discover and describe additional examples of these systems, a mechanistic understanding of how they achieve behavioral manipulation in their animal hosts has been limited by the lack of a genetically-manipulable model system. We believe that the combination presented here - a strain of *E. muscae* that infects wild *Drosophila*, protocols for propagating this strain in lab-reared flies, and the robust induction of behaviors in the laboratory - has great potential as a model system that allows us to wield the tools of modern molecular genetics and neuroscience to describe the molecular mechanisms that underlie at least one example of microbial manipulation of animal behavior.

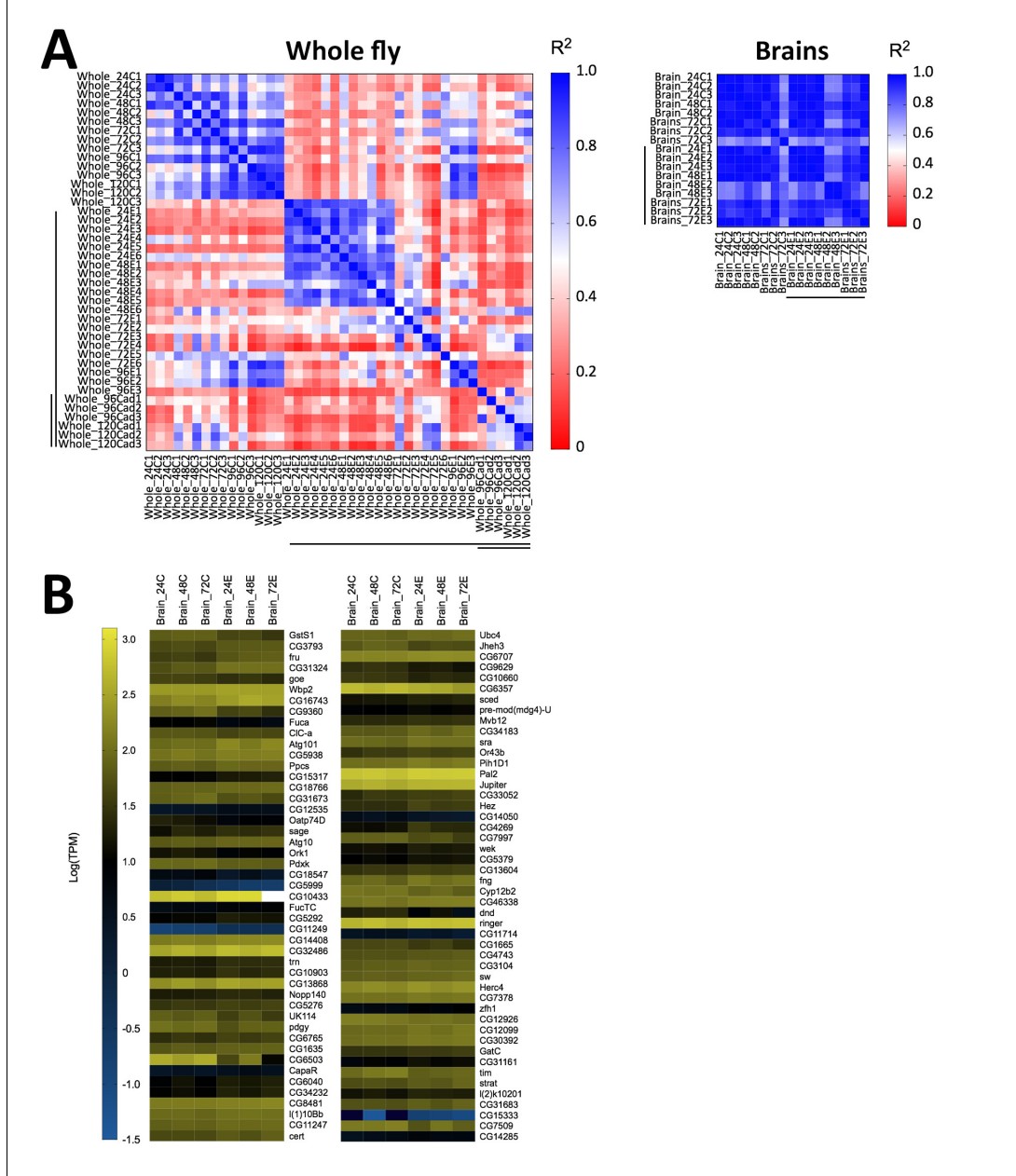

**Figure 8.** Host gene expression in the brain is minimally altered over the first 72 hr of *E. muscae* 'Berkeley' infection. (A) Left) All pairwise linear correlations between samples from *E. muscae* 'Berkeley'- infected whole fly RNAseq time course. Right) All pairwise linear correlations between samples from *E. muscae* 'Berkeley'-infected dissected brain pilot RNAseq time course. Samples are named in the following format: Tissue_HourTreatmentReplicate, with 'C' indicating controls, 'E' indicating exposed flies living at the time sampled and 'cad' indicates that the fly had been killed by *E. muscae* 'Berkeley' before sampling. For example, 'Whole_24C1' indicates the first replicate control sample (uninfected fly) of a whole animal taken at 24 hr after mock exposure. Living, exposed samples are denoted on each axis with a black bar; infected cadaver samples are denoted with a double black bar. (B) Heatmap showing all genes that are differentially expressed (p value < 0.001) between controls (uninfected) and exposed flies at 24–72 hr, as determined by ANOVA. Genes are shown in order of increasing p-value from top left corner to bottom right corner.

DOI: https://doi.org/10.7554/eLife.34414.027

The following figure supplement is available for figure 8:

**Figure supplement 1.** *E. muscae* 'Berkeley' transcripts detected in dissected brains.

DOI: https://doi.org/10.7554/eLife.34414.028

### *E. muscae* in wild drosophilids

Though to our knowledge we are the first to study a naturally *Drosophila*-infecting strain of *E. muscae* in the laboratory, we are not the first to encounter *E. muscae* circulating in wild *Drosophila*. In 1927, Goldstein reported finding *Empusa muscae* (now *Entomophthora muscae*)-infected cadavers of *Drosophila repleta* as well as *Musca domestica* at Columbia University in New York state, stating that an epidemic of *E. muscae* had been observed for the previous four years in this location (*Goldstein, 1927*). In 1969, Turian and Wuest reported observing *E. muscae*-infected cadavers of wild *Drosophila hydei* in a rotting fruit bait in Geneva, Switzerland (*Turian and Wuest, 1969*). In 2002, Keller et al. reported morphological parameters for an *E. muscae* strain (putatively identified as *E. ferdinandii*, a member of the *E. muscae* species complex) infecting *Drosophila spp* in Switzerland (*Keller, 2002*).

It remains unclear both (1) if the strain(s) or species infecting *Drosophila spp* are distinct from those that infect other fly species and (2) the degree of specificity for *Drosophila spp* over other dipterans. For example, we have observed that *E. muscae* 'Berkeley'-infected *D. melanogaster* cadavers are capable of infecting *M. domestica* in the laboratory, but it is unclear if this infection would occur in the wild (*Figure 3—figure supplement 1*). Whether infection can occur naturally would depend on the ecology of the two different host species (i.e. if they interact frequently enough to expect the exchange of *E. muscae* infection) and the efficiency of infecting each species under natural conditions. Findings from previous work examining host specificity of other *E. muscae* strains have not always been consistent. While some have found evidence for differences in infection efficiency between and local adaptation to varying hosts (*Gryganskyi et al., 2013*; *Mullens, 1989*), others have demonstrated the capacity of a given *E. muscae* isolate to infect a variety of host taxa (*Steinkraus and Kramer, 1987*). Our understanding of strain diversity and host specificity would greatly benefit from the collection of more molecular and ecological data for *E. musace* complex species and infected hosts.

### The progression of *E. muscae* 'Berkeley' infection in lab-reared *D. melanogaster*

Taken together, our RNAseq and histology time course data describe the typical progression of *E. muscae* infection in lab-reared *D. melanogaster*. At 24 hr after exposure, flies show a robust antifungal immune response, though the fungus is nearly undetectable within the fly by histology, indicating that it is at low titer. At 48 hr, the fungus has begun to adopt the morphology which it will assume until killing the host. Fungus is observable within the host brain and VNC, though overall fungal titer is still quite low. As the abdomen is the most likely point of initial entry for the fungus (it is the biggest target for the fungus to hit), we suspect that the fungus has travelled from the point of entry to the CNS, indicating tropism for neural tissue early in the infection. Elements of the host's immune response are still activated. At 72 hr, fungus was apparent throughout the body cavity, in the thorax (between muscle fibers), abdomen (surrounding but not invading the gut and gonads) and also in the limbs and halteres. The fat body is significantly depleted by this point; the host's dampened metabolism suggests an internal starvation state. At 96 hr, if the fly has not succumbed to infection (i.e. there are still energy reserves available to the fungus), fungal titer will continue to increase and the fat body will continue to be depleted. Two-three hours after death, flies that have been killed by the fungus show no intact abdominal organs and nervous systems that are being broken down.

While the trajectory of infection is consistent, it is important to recognize that just because two animals were exposed to infectious cadavers at the same time point does not guarantee that they were infected at exactly the same time (since sporulation occurs over a window of several hours) or that, once infected, that these two animals will progress through infection identically. In our RNAseq data, we noticed that the host gene expression in exposed animals at 24 and 48 hr tended to be more variable than those for 72 hr. We imagine that this was due to chance, that we picked animals that were at similar points in infection at 72 hr, whereas we picked animals that were more variable at other time points. This may at least partly explain why we observed so much differential expression at the 72 hr time point in exposed versus controls whereas less was observed at earlier time points, especially at 48 hr (*Figure 4—figure supplement 2*). It is likely that several factors play into whether or not an infection succeeds and how quickly it progresses (e.g. initial exposure titer, size of host, nutritional status of host etc.). Thus, future work should consider determining a metric to gauge

progress of infection so that similarly-progressed samples can be compared and, where possible, use larger sample sizes to mitigate within-group variation.

## E. muscae 'Berkeley' infection and host immune response

The entomopathogen community has maintained that *E. muscae* evades the host response by growing protoplastically (i.e. without a cell wall, components of which would be recognized and targeted by the host immune system). In both the gypsy moth and the greater wax moth, it has been shown that the host immune cells recognize walled *Entomophaga* fungal cells, but there is little cellular response to protoplasts (*Butt et al., 1996*; *Beauvais et al., 1989*). Based on these findings, it has been posited that the host does not detect the ever-increasing fungal mass within until the end of infection when the fungus puts on a cell wall that contains epitopes that the host can recognize (*Boomsma et al., 2014*; *De Fine Licht et al., 2017*). As a result of ostensibly evading the immune system, it has also been hypothesized that *E. muscae* does not generate toxins, as it would have no incentive to do so in the absence of attack by the host (*Boomsma et al., 2014*).

Our data show that there is a robust initial response to *E. muscae* 'Berkeley' exposure. Many of the immune genes that are induced with *E. muscae* 'Berkeley' have also been observed to be induced by exposure to other, more generalist fungal pathogens (e.g. *Beauveria bassiana*, *Metarhizium anisopliae* [*Valanne et al., 2011*; *Levitin and Whiteway, 2008*]). These data clearly indicate that the host detects an invader early on in infection. Furthermore, there is a detectable immune response through the length of infection (*Figure 6*, *Figure 4—figure supplement 3*), though at this point we cannot say if this response is a slow disengagement of the initial response or stimulated de novo by the growing fungus.

Interestingly, the living animals sampled at 96 hr for RNAseq are inconsistent in their host transcriptional immune response: two of the three animals more closely resemble control animals than infected animals in host transcription (*Figure 4D*, *Figure 4—figure supplement 3*). At least two scenarios could explain this observation. It is possible that there was a delay in the course of infection compared to contemporaneous samples or that both of these animals were in the process of recovering from infection (i.e. the immune system was effectively combating the fungus). The proportion of fungal reads present in these samples is lower than what would be expected for late time points, which is consistent with either scenario. At this point we simply do not know if every instance of a spore hitting a fly leads to a productive fungal infection. There is some evidence to the contrary: we have consistently observed that some highly- exposed flies die prematurely. These animals are generally smaller than others in the vial and are often covered in spores. This could indicate that getting hit by too many spores (an unlikely outcome in the natural world) leads to an overwhelmed fly (e.g. overactive immune system or accelerated fungal growth) that dies before being manipulated. These flies do not sporulate, though it is possible that they do produce resting spores. On the other hand, we have observed that survival of exposed flies is substantially increased when flies are exposed to small quantities of antifungal. This indicates that there are ways of either halting or slowing an infection, though whether the fly's immune system is generally capable of doing this is unknown.

It was previously proposed that this fly's immune system is relatively quiescent during protoplastic fungal growth then becomes overwhelmed by pathogen biomass as the fungus puts on a cell wall in preparation to leave the host (*Boomsma et al., 2014*). If this were the case, we would expect to observe a sizeable increase in immune gene expression at or immediately preceding the end of life. Five of our six cadavers sampled for transcriptomics have similar levels of immune gene transcripts compared to *E. muscae* 'Berkeley'-exposed animals sampled at 72–96 hr, which is inconsistent with this trend, however, the sixth cadaver exhibits higher expression of antifungal peptides (Drosomycin, Attacin-A, B and C, CecropinA1 and A2, DiptericinA-B and Metchkinowin), edin (elevated during infection) and peptidoglycan recognition protein SA. It could be the case that the immune system is overwhelmed in all animals immediately preceding death; we could have sampled too late to observe it in the other five samples but were able to see it in one sample that was late to respond. However, one can imagine that in the presence of copious fungal epitopes the fly immune system would continue to be highly engaged until death, not return to levels comparable to earlier time points (i.e. 72 hr), as seen in the majority of sampled cadavers. While the majority of our data is inconsistent with the notion that the fly's immune system is overwhelmed immediately preceding death, we cannot formally exclude this possibility given our small sample size (six individuals).

## Why is *E. muscae* 'Berkeley' in the brain?

Our work demonstrated that *E. muscae* 'Berkeley' is present in the nervous system system relatively early in infection, just 48 hr following exposure. The invasion of the nervous system by *E. muscae* 'Berkeley' grants the fungus direct access to host neurons and may be mechanistically important for achieving behavioral manipulation of the host fly. However, we should be careful to consider any and all possible ways *E. muscae* 'Berkeley' could alter host behavior before jumping to this conclusion.

We can imagine four general mechanisms by which *E. muscae* is able to achieve behavioral manipulation. The fungus could invade the nervous system in order to localize adjacent to and impinge on the activity of particular neurons through chemical or physical means. However, we are skeptical that this is the case as our observations do not support specific localization of the fungus in the CNS.

A second possibility is that the fungus invades the nervous system in order to gain access to either a particular group or groups of neurons or all neurons generally, but does not localize within the CNS in a stereotyped manner. Rather it is sufficient that it has crossed the blood brain barrier, which insulates the nervous system from the activities in the hemolymph and allows for the selective transport of compounds to and from the hemolymph, allowing the fungus to modulate the activity of neurons by secreting compounds that diffuse throughout the CNS. The secreted compounds could be specific, only altering the activity of a subset of susceptible neurons, or could be more general, changing activity over many or all neurons. This is somewhat akin to the mechanism by which jewel wasps (*Ampulex compressa*) alter cockroach behavior - after an initial injection of a chemical cocktail that sedates the cockroach, the wasp then injects toxins into the brain, which decreases neural activity and renders the cockroach unlikely to spontaneously initiate walking though still capable of walking when lead by the wasp (*Haspel et al., 2003*; *Gal and Libersat, 2010*). Interestingly, another Entomophthoralean fungus, *Strongwellsea magna,* is also known to invade the nervous system of its lesser house fly host (*Fannia canicularis*) during infection (*Humber, 1976*). In this case, the author proposed that this did not have consequences for behavior.

A third possibility is that the fungus does not need to invade the nervous system in order to change the host's behavior. The fungus could be secreting a compound into the hemolymph that is capable of crossing the blood brain barrier and altering neuronal activity. Alternatively, the fungus could be secreting a compound into the hemolymph that changes the host's internal state (either directly or by leading the host to respond in a way that causes the internal state to change), which leads the animal to respond by executing one or more of the end-of-life behaviors. *Ophiocordyceps unilateralis* likely employs this strategy in its carpenter ant host, *Camponotus castaneus* - the fungus never invades the nervous tissue but still induces clear behavioral manipulations, which are almost certainly underpinned by changes in host neurobiology (*Fredericksen et al., 2017*).

Lastly, it's possible that the fungus does not secrete compounds to induce these behaviors. Instead, it could be that these behaviors are naturally elicited in response to large-scale tissue destruction. While we believe this last scenario to be highly unlikely, it cannot yet be ruled out.

For these last two proposed mechanisms, the fungus would not need to invade the CNS in order to affect behavior. In these cases, the fungus could be invading the CNS as a means of escaping immune surveillance. By establishing a reservoir in the CNS, the fungus could replenish dying cells in the hemolymph in order to ensure that the infection took hold. Alternatively, the fungus could invade the CNS because it provides a rich, nutritive substrate to sustain the fungus. This scenario is inconsistent with our histological data both from flies that are not executing end-of-life behaviors (*Figure 6*) and flies that are executing end-of-life behaviors show that the brain is largely intact, indicating that the fungus abstains from consuming these tissues until host death.

Transcriptomic analysis of dissected brains from exposed females at 24, 48 and 72 hr with confirmed *E. muscae* 'Berkeley' infections showed only subtle changes in host gene expression compared to uninfected controls (*Figure 8*). We propose that single-cell transcriptomic profiling of brains from infected animals actively exhibiting fungal-induced behaviors could be a means to gain insight into which specific neuronal populations are targeted by the fungus and the molecular mechanisms within these cells that mediate fungal manipulation. Consistent with our histological experiments, our transcriptomic data showed that *E. muscae* 'Berkeley' is present in the brain and that,

taking *E. muscae* 'Berkeley' reads as a proxy, fungal titer increases with the progression of infection (*Figure 8—figure supplement 1*).

## Behavior and beyond: the utility of the *E. muscae* 'Berkeley'-*D. melanogaster* system

The past decade has seen an explosion of tools for characterizing and manipulating the nervous system of *D. melanogaster*, including a catalog of the types and corresponding expression patterns of its approximately 100,000 neurons, a complete map of connections in the brain, reagents for conditionally activating or inactivating specific sets of neurons as well as purifying these cells, and methods for automatically tracking and classifying behaviors in populations. Our development of a robust system of microbially induced behavior manipulation in *D. melanogaster* will allow us, and we hope many others, to leverage the powerful molecular and neurobiological toolkit of *D. melanogaster* to explore the molecular basis of this fascinating but still mysterious biological phenomenon.

## Materials and methods

### Fly husbandry

Healthy wild-type, CantonS Wolbachia-free (WF) D. melanogaster were reared on Koshland diet (0.68% agar, 6.68% cornmeal, 2.7% yeast, 1.6% sucrose, 0.75% sodium tartrate tetrahydrate, 5.6 mM CaCl2, 8.2% molasses, 0.09% tegosept, 0.77% ethanol, 0.46% propionic acid) supplemented with activated dry yeast pellets (Red Star) at 21°C on a 12:12 light:dark photoperiod. Any time the photoperiod needed to be adjusted, flies were reared from third instar or earlier on the desired photoperiod to ensure that emerging adults were properly entrained.

### Fendel tending

Wild fruit flies were caught by directly aspirating from an uncovered plastic dishwashing pan (aka the 'fendel') that was baited with quartered organic watermelon and an assortment of other organic fruits. Aspirated flies were transferred onto Koshland diet and housed at ambient temperature and ambient humidity. Baiting and capture was performed in the spring through early fall of 2015 at a personal residence in Berkeley, CA.

### PCR genotyping

DNA was extracted from individual cadavers or 1.5 mL of in vitro culture using the QIAamp Micro Kit (QIAGEN) following the tissue protocol. These DNA preparations were used to amplify the desired sequences. Entomophthora-specific ITS primers (emITS: emITS-1 5'- TGGTAGAGAATGA TGGCTGTTG-3', emITS-4 5'- GCCTCTATGCCTAATTGCCTTT-3') or fungal- specific large subunit primers (LSU: LR3-1 5'- GGTCCGTGTTTCAAGAC-3', LR0R-4 5'- GTACCCGCTGAACTTAAGC-3') were used to genotype Entomophthora (*James et al., 2006*); cytochrome oxidase II primers (tLEU: 5' ATGGCAGATTAGTGCAATGG 3' and tLYS: 5' GTTTAAGAGACCAGTACTTG 3') were used to genotype infected Drosophila hosts *Liu and Beckenbach (1992)*. Each reaction was performed using GoTaq 2x colorless mastermix (Promega) with 800 nM of each forward and reverse primer with the following thermocycling conditions: 95°C for 5 min followed by 35 iterations of 95°C for 30 s, 51°C for 30 s then 72°C for 1 min/kb then 72°C for an additional 10 min. Reactions were checked by gel electrophoresis in 1% agarose. Successful reactions were prepared for sequencing using ExoSap-IT (Affymetrix) per manufacturer's instructions and submitted with each amplification primer for Sanger sequencing. Assembly of forward and reverse reads was attempted to generate a consensus sequence using Seqman Pro (DNA Lasergene v.10). Sequences were searched against the BLAST NT database using blastn.

### Isolating *E. muscae* 'Berkeley' in vitro

To grow E. muscae 'Berkeley' in vitro, first spores were collected using the ascending conidia collection method (i.e. by placing a fresh cadaver in the bottom of a sterile petri dish and allowing the cadaver to sporulate overnight) (*Hajek et al., 2012*). The following morning, the lid of the dish was rinsed with 10 mL of Grace's insect medium (1x) supplemented with L-glutamine, 3.33 g/L lactalbumin hydrolysate, 3.33 g/L yeastolate (ThermoFisher Scientitfic # 11605–094) containing 5% fetal

bovine serum (Invitrogen) and transferred to a vented, treated T25 tissue culture flask (Corning # 10-126-10) using sterile technique. The culture was then incubated at room temperature in the dark until growth was apparent (non- homogenous white spheres floating in the medium). The culture was genotyped with emITS and LSU primers to confirm that it was *E. muscae* and was an identical match to the cadaver that produced the spores which started the culture ('Fendel mama') at these loci. The culture was periodically examined at 100-400x on a compound microscope to confirm proper morphology and absence of contamination. The isolate is deposited in the USDA Agriculture Research Service Collection of Entomopathogenic Fungal Cultures (ARSEF) under accession number 13514.

## Isolating and optimizing in vivo *E. muscae* 'Berkeley' infection

Wild fruit flies sampled from the fendel and housed on Koshland food were monitored nightly for two weeks after capture for deaths due to *E. muscae* 'Berkeley'. Freshly killed cadavers were separated from their living conspecifics by briefly anesthetizing live flies via cold anesthesia (incubating 2–3 min in a residential freezer). Fresh cadavers (anywhere from 1 to 10, depending on availability) were placed on chunks of organic banana in a wide Drosovial with kimwipes to sop up excess moisture. Approximately, 50 healthy flies were then transferred onto the prepared vial by flipping (no anesthesia was used). The plug of the vial was pushed down to confine the flies within a few centimeters to improve the likelihood that they would encounter flying spores. Leaving the exposed flies with the spent cadavers was initially problematic as we were working without access to anesthesia or a microscope and had to identify new cadavers by naked eye. Additionally, the raw banana began to ferment and break down, leading to excess moisture which was prematurely killing some of our exposed flies. To avoid these issues, the exposed flies were transferred to a new banana/kimwipe vial after the first 48 hr. This was done by allowing the living flies to climb into an empty 'holding' vial then flipping them onto the fresh vial. The flies were monitored daily for deaths; cadavers were removed after allowing healthy flies to climb into a 'holding' vial and used to start new exposure vials.

## In vivo propagation of *E. muscae* 'Berkeley' infection

Cadavers are collected daily 2–5 hr after the end of the 12 hr light cycle from exposure vials that are between 96 and 168 hr (4 and 7 days) old. All flies that will die on this day because of *E. muscae* 'Berkeley' infection will be dead by this time and will be extremely swollen with fungal growth, making them obvious among the living flies. $CO_2$ is used to anesthetize the living flies in exposure vials and collect cadavers, which are placed in a petri dish with a piece of Whatman paper wetted with deionized water to mitigate static. Molten cadaver embedding medium is preparing by microwaving solidified AS solution (1.5% agar, 10% sucrose) and poured into a clean 100 × 15 mm petri dish just enough to cover the bottom of the dish. As soon as the agar has set, six cadavers are embedded head first in a circle of diameter <2 cm with their wings on the outside of the circle. The wings are pressed into the agar to ensure they do not intercept any launched conidia. The agar is allowed to completely set before continuing. The cadaver circle is cut out from the set agar by using an empty, wide-mouth Drosophila vial (FlyStuff). The agar disc containing the fly circle is then transferred, cadaver-side up, into an empty, wide-mouth Drosophila vial. A ruler is used to mark 2 cm above the surface of the agar. CantonS flies reared on the same 12 hr light cycle on Koshland medium are collected from eclosing vials using $CO_2$ anesthesia. Fifty healthy flies are added to the vial container the agar disc with cadavers and tapped down until all flies are under the 2 cm mark. A Droso-plug (Fly-Stuff) is pushed into the vial such that its bottom is flush with the 2 cm mark. The vials are incubated for the first 24 hr at 18°C in a humid chamber (~95% humidity, 2L plastic beaker lined at the bottom with wetted paper towels or kimwipes and covered with foil), to encourage sporulation. After 24 hr, the Droso-plug is lifted to relieve the confinement of the flies and the vial is moved to a 21°C incubator (~60% humidity). After 48 hr, the exposed flies are transferred onto GB+ medium (40% organic banana [w/v], 2% agar, 0.3% propionic acid) without anesthesia; incubation continues at 21°C. Cadavers are collected daily 2–45 hr after the end of the 12 hr light cycle from exposure vials that are between 96 and 168 hr (4 and 7 days) old. The process is repeated daily to supply cadavers for experiments and ensure the infection is maintained.

## Photography and videography

Pictures and videos of dead and dying flies (extending probosces and raising wings) were taken with a Nexus 5x (Google) or iPod Touch (Apple) aided by attaching a macrolens (Luxsure) to the device camera lens or by aligning the device camera lens with the eyepiece of a dissecting scope. Images were taken under ambient light, humidity and temperature.

Time lapse microscopy was taken via a USB microscope (DinoLite Digital Microscope Pro) using DinoLite software v1.12 (*Figure 3A,B*) or on a Nikon 80i compound microscope equipped with a Hamamatsu black and white camera (C11440) using MetaMorph software (v. 7.8.00, Molecular Devices) (*Figure 3C,D*). Each time lapse consists of images collected once a minute for the indicated duration. Images were taken under ambient temperature and humidity.

High-speed videos (18,000–54,000 fps) were filmed with a 5x objective on a Axiovert 200 microscope (Zeiss) equipped with an Photron Fastcam 1024PCI. Cadavers were mounted in 1.5% agar on a glass slide and arranged such that forming conidiophores and conidia were visible (for spore ejection) or such that cadavers sporulated onto a #1 glass coverslip in the plane of the camera (for spore landing). Video was captured via Photron Fastcam Viewer software, running at the indicated frames per second in end trigger mode (new frames were continually captured and old frames dumped until the user triggered the filming to stop). Spores or landing sites were manually watched until a spore disappeared or appeared, then video was stopped and last ~10 s of footage were manually searched for the spore launching or landing event.

## Circadian timing of death

CantonS WT flies were reared on a 12:12 light cycle (photophase 1 am – 1 pm or 7 pm -7 am PDT, as indicated). For experiments run in complete darkness, flies were exposed to *E. muscae* 'Berkeley' via the anesthesia-dependent protocol with 25 adults exposed per vial instead of the standard 50. All exposure vials were set up before the utilized cadavers sporulated (within 2 hr of the light-dark transition). Exposure vials were housed in a humid chamber in a dark 21°C incubator wrapped in blackout cloth for approximately 24 hr before loading into Drosophila activity monitors (DAMs, Trikinetics). Before loading flies, DAM tubes (5 x 65 mm, polycarbonate) were prepared containing such that one end of the tube held approximately one cm of 1.5% agar, 10% sucrose and was wrapped in parafilm to prevent drying out.

To load the flies, all accessible light sources were disabled before removing the humid chamber from 21°C and placing on the bench at RT. Vials of flies were kept under a foil-lined box as they waited to be processed. One vial at a time was retrieved from the box, knocked out with $CO_2$ under a dissecting scope whose LED light ring was covered with a red gel filter (Neewer), sorted by sex and loaded into individual DAM tubes with the aid of a red LED flashlight (KMD Aero) before capping each tube with an autoclaved cotton plug. For each vial, eight females and eight males were chosen for loading into DAM vials. Unexposed flies (i.e. controls) were always processed before proceeding to exposed flies. $CO_2$ pad was wiped down with 70% ethanol between vial types to present cross-contamination. DAMs were loaded from bottom to top row, filling a row and securing each tube with rubber bands before proceeding to the next. Loaded tubes were kept under a foil-lined box to prevent light exposure. When all loading was finished, DAMs were covered with blackout cloth and transported back to the 21°C incubator. There they were attached to the DAM interface unit and recording began, binning counts in 30 s intervals. Recording continued for ~170 hr until the experiment was stopped.

Like with loading, experiments were stopped by first disabling all light sources, then carefully disconnecting and removing DAMs from the 21°C incubator to not disturb adjacent experiments, and sealing incubator before turning on the overhead lights for manual inspection. Each DAM tube was inspected to see if the fly was dead or alive. If dead, the tube was inspected for evidence of sporulation to determine if the cause of death was patent *E. muscae* infection. For experiment run on a 12:12 light cycle, flies were exposed as above but without concern for light contamination; resultant DAMs were housed on a 12:12 light cycle for the duration of the experiment. Each channel was checked daily following sundown to see which flies had died within the previous 24 hr.

DAM data were processed in Python to determine time of last movement (accurate to 30 s) and to plot movements over time. For each channel, the reported time of last movement was manually cross- checked using the plot of activity data. In cases where there was an erroneous movement (i.e.

a signal occurring more than 24 hr after the fly's last movement), the time of last death was manually re-assigned. For data visualization, data were binned into 15–30 min intervals and the average movements of unexposed animals (controls), exposed or entrained light cycle of exposed flies and cadavers and the time of last movement for each observed cadaver were plotted in Prism (Graph-Pad). There were no obvious differences between male and female activity for the unexposed animals so sexes were combined for data analysis.

## Collection and staining of primary conidia

Three to six fresh cadavers (i.e. those who had not yet sporulated) were collected from exposure vials using the anesthesia dependent methods detailed above. Sporulation chambers were prepared as follows: a small piece of Whatman paper was placed in the base of a small petri dish (60 x 15 mm) and wetted with DI water. A bloated cadaver was chosen for each chamber and its wings were removed. The cadaver was placed in the middle of the Whatman paper and the chamber was topped with a custom, 3D-printed top that included a square opening slightly smaller than a standard $22 \times 22$ mm coverslip. The top and bottom were sealed using parafilm and a new coverslip was placed over the opening. Cadavers were left in the chambers at room temperature to sporulate. Coverslips were changed every 30 min to 1 hr, as needed, and promptly stained for microscopy by applying Hoechst (1 µg/mL). Spores were imaged on a compound microscope at 40x for measuring conidia attributes; exact distances were determined by calibration with a 0.01 mm micrometer (OMAX). For each attribute (number of conidia, length and width of conidia, diameter of nuclei), at least 50 different primary conidia were counted from three different cadavers.

## RNA experiments

RNA was prepared from each thawed sample by homogenizing with an RNase-free pestle (Kimble Chase), washing the pestle with 750 µL Trizol, then proceeding using the manufacturer's protocol. For reverse-transcription PCR (RT-PCR) and transcriptomic experiments, three mock vials and three exposure vials were started with 25 CantonS WF flies 0–1 days old (RT-PCR, whole flies) or 1–2 days old (dissected brains) with either zero (mock) or six (exposure) cadavers embedded in AS. Flies were incubated for the first 24 hr at 18°C confined to 2 cm with cadavers, then moved to 21°C where the confinement was relieved. Flies were transferred to GB +at 48 hr where they continued to be housed at 21°C. (Vials were sampled every 24 hr for 72 (dissected brains), 96 (RT-PCR) or 120 (whole flies) hours by anesthetizing the entire vial with $CO_2$. Exposed animals were preferentially selected based on evidence of contact with fungus (e.g. spores or melanization responses thereto visible on cuticle). Sampling for each time point consistently occurring between 2–3 hr following the light-dark transition. Before sampling, all equipment used to manipulate flies (e.g. $CO_2$ pad, forceps etc.) were treated with 10% bleach, wiped with DI water then sprayed with 70% ethanol. All materials that handled flies ($CO_2$ gun, pad, forceps) were treated with 10% bleach and rinsed with DI water between sampling exposure vials. Control vials were always sampled first. Sampled material (either whole fly or dissect brain) was immersed in 250 µL Trizol then immediately flash frozen with liquid nitrogen. Samples were stored at −80C until extraction.

## RT-PCR

One control female (24–96 hr) and two infected females (24–72 hr) or one fresh cadaver (96 hr) from each vial were collected as described above. RNA was prepared from each thawed sample by homogenizing with an RNase-free pestle (Kimble Chase), washing the pestle with 750 µL Trizol, then proceeding using the manufacturer's protocol. RNA was then treated with Turbo DNase (Thermo-Scientific) per the manufacturer's protocol and quantified using a Qubit Fluorometer (Qubit RNA HS assay kit, ThermoFisher Scientific). For each sample, 1 µL or 160 ng of DNase-treated RNA, whichever was more, was added to a new, nonstick tube and mixed with two pmol primer emITS1, 770 nM dNTPs in a final volume of 13 µL. The reaction was incubated at 65°C for 5 min then incubated on ice for at least 1 min before proceeding. To the mixture was added 5x First Strand Buffer (1x final, ThermoFisher Scientific), 100 mM DTT (5 mM final, ThermoFisher Scientific), 1 µL RNaseOUT (ThermoFisher Scientific) then 200 units of SuperScript III RT (ThermoFisher Scientific). After thorough mixing, each tube was incubated at 55°C for 60 min to reverse transcribe then 70°C for 15 min to heat kill the transcriptase. To amplify E.muscae-specific cDNA, 2 µL of the reverse transcription

reaction was mixed with GoTaq 2x colorless mastermix (1x final, Promega) and 500 nM each primers emITS1 and emITS4 (5'- GCCTCTATGCCTAATTGCCTTT-3') then run on a thermocycler with the following settings: 95°C for 5 min followed by 35 iterations of 95°C for 30 s, 61°C for 30 s then 72°C for 30 s then 72°C for an additional 10 min. Four µL of each reaction was analyzed by gel electrophoresis in 1% agarose.

## Whole fly in vivo RNAseq time course

One control female (24–120 hr) and two infected females (24–72 hr), one infected female and one cadaver (96 hr) or one cadaver (120 hr) from each vial were collected as described above. RNA was prepared from each thawed sample by homogenizing with an RNase-free pestle (Kimble Chase), washing the pestle with 750 µL Trizol, then proceeding using the manufacturer's protocol. RNA was quantified using a Qubit Fluorometer (Qubit RNA HS assay kit, ThermoFisher Scientific) and quality was checked by running on a RNA 6000 Pico chip on a Bioanalyzer 2100 (Agilent Technologies). High quality RNA was then treated with Turbo DNase (ThermoScientific) per the manufacturer's protocol. RNAseq libraries were prepared with the TruSeq RNA v2 kit (Illumina) using 500 ng of input RNA per sample. Samples were multiplexed 21 samples to a lane and sequenced using 100 bp paired-end reads on a HiSeq 4000 at the QB3 Vincent J. Coates Genomic Sequencing Facility at UC Berkeley. Raw reads are available through NCBI's Gene Expression Omnibus (GEO) in series GSE111046.

## Dissected brain RNAseq

Brains were individually dissected and sampled from first three control and then three exposed females. Each animal was dissected in sterile 1x PBS in its own disposable dissection chamber (35 mm petri dish lined with 2–3% agar) and dissecting forceps were treated with 3.5% hydrogen peroxide then rinsed with sterile water between samples to prevent nucleic acid carryover. The body of each animal was saved and subjected to a DNA extraction using the manufacturer's provided protocol for the isolation of genomic DNA from tissues (QIAamp DNA Micro kit, QIAGEN) eluting in 20 µL of buffer AE. For each fly body, 1 µL was used to template a PCR reaction consisting of 12.5 µL GoTaq, 2 µL of each primer emITS1 and emITS4 (10 µM stocks), and 7.5 µL water for a final volume of 25 µL. Reactions were cycled with the following conditions: 95°C for 5 min followed by 35 cycles of 95°C for 30 s, 51°C for 30 s and 72°C for 1 min, then a final 10 min extension at 72°C. Reactions were analyzed via gel electrophoresis to confirm that all exposed animals had come into contact with *E. muscae* 'Berkeley' and that control animals were uninfected.

RNA was prepared from each thawed sample by homogenizing with an RNase-free pestle (Kimble Chase), washing the pestle with 750 µL Trizol, then proceeding using the manufacturer's protocol. RNA was quantified using a Qubit Fluorometer (Qubit RNA HS assay kit, ThermoFisher Scientific) and quality was checked by running on a RNA 6000 Pico chip on a Bioanalyzer 2100 (Agilent Technologies). One replicate control RNA sample for the 48 hr time point was lost prior to library preparation so was omitted. High quality RNA was then treated with Turbo DNase (ThermoScientific) per the manufacturer's protocol. RNAseq libraries were prepared with the TruSeq RNA v2 kit (Illumina) using all of the extracted RNA for each brain, 17–75 ng of input RNA per sample. Samples were multiplexed 17 samples to a lane in equimolar amounts and sequenced using 100 bp paired-end reads on a HiSeq 4000 at the QB3 Vincent J. Coates Genomic Sequencing Facility at UC Berkeley. Raw reads are available through NCBI's GEO in series GSE111046.

## *E. muscae* 'Berkeley' de novo transcriptome assembly

An initial reference (Emus-Ref1) was assembled from reads from exposed in vivo time course samples that had first failed to align as pairs to the *D. melanogaster* transcriptome (r6.11, HiSat2) then failed to align as singletons to the *D. melanogaster* genome (r.611, bowtie2) using TRINITY with the developer's recommended settings (*Grabherr et al., 2011*). After assembly, all in vivo time course reads were aligned to Emus-Ref1 to assess contamination of non-*E. muscae* sequences as estimated by percentage of reads from control (unexposed) samples that aligned to the reference. All Emus-Ref1 transcripts were searched using blastn for homology (evalue 1e-40 or smaller) to organisms not annotated with Blast names 'fungi' or 'chytrid' or Blast species name 'Twyford virus'. These transcripts were removed to generate Emus-Ref2. All in vivo time course reads were aligned to Emus-

Ref2 to assess contamination of non-*E. muscae* sequences. Transcripts that were not expressed by any sample (TPM = 0) or where TPM of uninfected samples accounted for more than 10% of TPM summed across all samples were removed to generate Emus-Ref3. All in vivo time course reads were aligned to Emus-Ref2 to assess contamination of non-*E. muscae* sequences. Transcriptome completeness was estimated by BUSCO v1.1 analysis using the fungal reference set (1438 BUSCOs).

## RNAseq data analysis

To calculate gene expression, we aligned reads sequentially to the following references, with only reads not aligned to the previous genome used as input for the next genome: *D. melanogaster* mRNAs, *D. melanogaster* non-coding RNAs, unannotated *D. melanogaster* transcripts, the *D. melanogaster* genome version 6.21, a collection of microbial genomes, the coliphage phiX174 genome (used in calibrating Illumina sequencers and a common contaminant of sequencing data), the genomes of several viruses, unannotated transcripts not present in current versions of either the *D. melanogaster* or *E. muscae* genomes, Emus-Ref3, annotated transcripts from the draft version of the *E. muscae* genome, unannotated *E. muscae* transcripts, and the draft version of the *E. muscae* genome.

The *D. melanogaster* RNAs and genome were version 6.21 downloaded from www.flybase.org. The phiX174 genome is GenBank accession AF176034.1. The microbial genomes, chosen because of the presence of reads aligning to these or closely related genomes in draft transcriptomes, were *Pichia kudriavzevii* (GenBank accessions CP028773.1-CP028778.1), *Saccharomyces cerevisiae* (strain S288C) downloaded from www.yeastgenome.org, *Gluconobacter albidus* (GenBank accession CP014689.1), *Gluconobacter oxydans* (GenBank accession LT900338.1), *Acetobacter pasteurianus* (GenBank accession AP014883.1), and *Leuconostoc pseudomesenteroides* (GenBank accession AM773723.1). The viruses used were Noravirus (GenBank accession JX220408.1) and a previously undescribed iflavirus. We used draft versions of the *E. muscae* genome and genome annotation available through NCBI (PRJNA479887). Prior to alignment we masked both Emus-Ref3 and genome annotated *E. muscae* transcripts by aligning using blastn against the *D. melanogaster* genome and converting any region in either *E. muscae* transcriptome with hits with e-value <1e-10 to N. Unannotated transcripts were identified by assembling reads from 72 hr infected samples from the time course experiment using Trinity (default parameters). Resulting transcripts large than 300 bp were aligned using blastn (default parameters) to the genomes of *D. melanogaster* and *E. muscae*. Any sequences with hits with e-value <1e-10 to either *D. melanogaster* or E.*E. muscae* (but not both) was considered an unannotated transcript from that species; all other sequences (those that aligned to neither or both genomes) were considered unassigned unannotated sequences.

Proportion of *D. melanogaster* aligned reads in *Figure 4A* were calculated by dividing the number of reads aligned to the *D. melanogaster* transcriptome reference by the sum of all reads aligned to the *D. melanogaster* transcriptome, the *E. muscae* 'Berkeley' transcriptome reference and annotated transcripts from the *E. muscae* 'Berkeley' genome. Proportion of *E. muscae* 'Berkeley' aligned reads were calculated as above, but using the sum of reads aligned to the *E. muscae* 'Berkeley' transcriptome and annotated transcripts from the *E. muscae* 'Berkeley' genome as the numerator. Transcript abundance (transcripts per million, TPM) was estimated from bowtie2 alignments (.bam files) using Salmon (*Patro et al., 2017*). Data were analyzed using hierarchical clustering by gene (Cluster 3.0), ANOVA between grouped treatments (scipy.stats) and GO term analysis (PANTHER [*Mi et al., 2016*]). Hierarchical clustering heatmaps were generated in Java TreeView; other data were plotted in matplotlib (Python), Prism (GraphPad) or Excel 2013 (Microsoft).

## Paraffin embedding and microtomy of whole flies

Two mock and two exposure vials were started daily for seven days each with 50 CantonS WF flies 0–1 days old with either 0 (mock) or 6 (exposure) cadavers embedded in AS. Flies were incubated for the first 24 hr at 18°C confined to 2 cm with cadavers, then moved to 21°C where the confinement was relieved. Flies were transferred to GB+ at 48 hr where they continued to be housed at 21°C. Vials were sampled every 24 hr via $CO_2$ anesthesia then infiltrated and embedded in paraffin. For detailed protocol, see dx.doi.org/10.17504/protocols.io.k5ecy3e. Briefly, flies were fixed 24–36 in ice-cold Carnoy's (6:3:1 ethanol:chloroform:glacial acetic acid) at 4°C. Samples were next dehydrated by stepping through a series of increasing ethanol concentrations Samples were then

transitioned into Histoclear (National Diagnostic) before slowly introducing Paraplast (Sigma). Samples were infiltrated with Paraplast for at least 84 hr at 60°C with gentle shaking before embedding in base molds with embedding rings (Thermo Scientific) and drying overnight. Samples were stored at room temperature until they were sectioned at 8 μm with an RM2255 microtome (Leica), applied to Polysine slides (ThermoFisher Scientific) and dried overnight at 42°C. Sections were stored at room temperature for up to three weeks before Safranin O/Fast Green FCF staining or up to one week before fluorescence in situ hybridization (FISH).

## Safranin O/Fast Green FCF staining of paraffin sections

Slide-mounted sections were dewaxed with two, 10 min changes of Histoclear then rehydrated to 70% ethanol with a decreasing ethanol series. Sections were then stained one-at-a-time following Johansen's Safranin and Fast Green protocol (*Ruzin, 1999*) then checked under a dissecting scope before mounting in DEPEX mounting medium (Electron Microscope Sciences) and drying overnight. Slides were imaged using a 20x objective with the Axio Scan.Z1 (Zeiss).

## Fluorescent in situ hybridization (FISH) of paraffin sections

Slide-mounted sections were dewaxed with two, 10 min changes of Histoclear then rehydrated to 70% ethanol with a decreasing ethanol series. Slides were incubated in 0.2 M HCl at 37°C for 45–60 min and rinsed in DI water before applying 80 μL of hybridization solution (20 mM Tris-HCl pH 8.0, 0.9 M NaCl, 0.01% sodium dodecyl sulphate, 30% formamide) containing 100 pmol/μL of an *E. muscae* 'Berkeley'-specific DNA probe (AlexaFluor633-5'-TGCTAAAACAGCACAGTT-3', ThermoFisher Scientific). Slides were incubated overnight in a humid chamber at room temperature. The following day, slides were briefly washed in 1x PBS with 0.3% Triton-X100, rinsed in 1x PBS and mounted in ProLong Gold with DAPI (ThermoFisher Scientific). Slides were cured for 24 hr before imaging on a LSM 800 confocal microscope (Zeiss) with 5x-40x air objectives.

## Acknowledgements

This research was supported by the MBE's Investigator Award from the Howard Hughes Medical Institute and CE's Graduate Research Fellowship from the National Science Foundation. This work used the Vincent J.Coates Genomics Sequencing Laboratory at UC Berkeley, supported by NIH S10 OD018174 Instrumentation Grant. The authors are grateful to Richard Humber whose expertise in all things *Entomophthora*, and his eagerness to educate us, was invaluable throughout, Michael Vahey and Brian Belardi in Dan Fletcher's group for assistance with high-speed videography, Steve Ruzin and Denise Schneides at the Berkeley Imaging Facility (College of Natural Resources, UC Berkeley) for their expertise and patience in microtomy and all things microscopy, Jen-Yi Lee for her guidance in using equipment in the Molecular Imaging Center (College of Natural Resources, UC Berkeley), Kristin Scott, Russell Vance, Damian Elias and Richard Calendar for advice, feedback and support during this project, and William Ludington for his reminder that 'fruit flies like a banana'. We are also grateful to Michael Bronski for his work in assembling the *E. muscae* 'Berkeley' genome (as well as introducing the Eisen Lab to the extraordinary world of entomopathogens), Jason Stajich for annotating our draft genome and Allison Quan for helpful comments and suggestions on the final draft. CE also acknowledges the support of Nora, Bruce and Kevin. All of the research described in this paper was funded by an HHMI Investigator award to MBE. CE was supported by a National Science Foundation Graduate Research Fellowship. CCM was supported by a National Science Foundation Postdoctoral Fellowship.

## Additional information

### Funding

| Funder | Author |
| --- | --- |
| Howard Hughes Medical Institute | Michael Eisen |
| National Science Foundation | Carolyn Elya<br>Ciera C Martinez |

The funders had no role in study design, data collection and interpretation, or the decision to submit the work for publication.

### Author contributions

Carolyn Elya, Conceptualization, Data curation, Formal analysis, Investigation, Visualization, Methodology, Writing—original draft, Writing—review and editing; Tin Ching Lok, Quinn E Spencer, Hayley McCausland, Investigation; Ciera C Martinez, Methodology; Michael Eisen, Supervision, Funding acquisition, Writing—original draft, Project administration, Writing—review and editing

### Author ORCIDs

Carolyn Elya (iD) http://orcid.org/0000-0002-9634-0303
Tin Ching Lok (iD) https://orcid.org/0000-0001-6388-5721
Hayley McCausland (iD) http://orcid.org/0000-0002-3177-2543
Michael Eisen (iD) https://orcid.org/0000-0002-7528-738X

### Decision letter and Author response

Decision letter https://doi.org/10.7554/eLife.34414.041
Author response https://doi.org/10.7554/eLife.34414.042

## Additional files

### Supplementary files

• Supplementary file 1. Summary of *E. muscae* infected *Drosophila* observed in California from 2014 until 2017.
DOI: https://doi.org/10.7554/eLife.34414.029

• Supplementary file 2. Morphology of primary conidia of *E. muscae* 'Berkeley' compared to other reported *E. muscae* strains.
DOI: https://doi.org/10.7554/eLife.34414.030

• Supplementary file 3. PANTHER functional classification of blastx results for fungal transcript groups as given in *Figure 4B*. Transcripts in groups i-iii were translated and queried against the Uni-Prot Swiss-Prot database using blastx. Uniprot identification numbers of transcripts with matches with an e- value less than 1e-40 were analyzed using PANTHER's functional classification analysis.
DOI: https://doi.org/10.7554/eLife.34414.031

• Supplementary file 4. GO term enrichments of host gene groups as given in *Figure 4D*. PANTHER GO-term analysis (complete biological process) for genes in Groups i-v (*Figure 4D*). FDR column lists the p-value from a Fisher's Exact test with false discovery rate correction for multiple testing.
DOI: https://doi.org/10.7554/eLife.34414.032

• Supplementary file 5. GO term enrichments of host gene groups as given in *Figure 4—figure supplement 2*. PANTHER GO-term analysis (complete biological process) for colored genes in *Figure 4—figure supplement 2*. FDR column lists the p-value from a Fisher's Exact test with false discovery rate correction for multiple testing.
DOI: https://doi.org/10.7554/eLife.34414.033

• Supplementary file 6. PANTHER functional classification of blastx results for fungal transcripts that are overexpressed in exposed compared to control samples at 72 hr post-exposure. Transcripts in were translated and queried against the UniProt Swiss-Prot database using blastx. Uniprot identification numbers of transcripts with matches with an e-value less than 1e-40 were analyzed using PANTHER's functional classification analysis.
DOI: https://doi.org/10.7554/eLife.34414.034

• Transparent reporting form
DOI: https://doi.org/10.7554/eLife.34414.035

### Data availability

Transcriptomic data have been deposited in GEO under accession code GSE111046. Genomic data have been deposited in NCBI under accession code PRJNA479887.

The following datasets were generated:

| Author(s) | Year | Dataset title | Dataset URL | Database, license, and accessibility information |
|---|---|---|---|---|
| Elya C, Eisen MB | 2018 | Transcriptomic response of Drosophila melanogaster whole bodies or dissected brains to Entomophthora muscae 'Berkeley' | https://www.ncbi.nlm.nih.gov/geo/query/acc.cgi?acc=GSE111046 | Publicly available at the NCBI Gene Expression Omnibus (accession no: GSE111046) |
| Bronski M, Elya C, Stajich J, Eisen MB | 2018 | Entomophthora muscae strain: Berkeley Genome sequencing and assembly | https://www.ebi.ac.uk/ena/data/view/PRJNA479887 | Publicly available at the European Nucleotide Archive (accession no: PRJNA479887) |

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
