## [Decision Letter]

Thank you for submitting your article "Robust manipulation of the behavior of *Drosophila melanogaster* by a fungal pathogen in the laboratory" for consideration by *eLife*. Your article has been reviewed by Patricia Wittkopp as the Senior Editor, a Reviewing Editor, and three reviewers. The following individual involved in review of your submission has agreed to reveal his identity: Brian P Lazzaro (Reviewer #1).

The reviewers have discussed the reviews with one another and the Reviewing Editor has drafted this decision to help you prepare a revised submission. There was a more extensive conversation than usual during the review process and all three reviewers provided detailed comments on the work, which is reflected in the length of the synthesized summary below.

The manuscript by Elya et al., describes a new strain of *Entomophthora muscae* that infects Drosophilids and manipulates the behavior of the infected host. The authors have made a considerable effort to culture the fungus and describe the host behaviors after infection. They additionally present a histological analysis of infection progression and some initial transcriptomics of both host and pathogen. All three reviewers agree that the system holds considerable potential for future study of the molecular and chemical mechanisms governing microbial control of host behaviors. Although the present manuscript is largely descriptive of the infection process the reviewers recognize the effort required for this initial characterization and feel that the prospect for future development of the system warrant potential publication of a revised manuscript in *eLife*. The transcriptomic analysis was considered to be weaker, and suggestions for improvement are detailed below.

The authors present transcriptional profiling of the host and pathogen during infection, but the fungal transcripts have no functional characterization so no biological interpretation can be made, and the host transcripts are primarily immune and metabolic responses that are largely concordant with what one would expect from a lethally infected insect. The former clearly stems from insufficient genomic characterization of *Entomophthora* and, while disappointing, cannot be addressed in this manuscript. The latter, however, almost certainly stems from the fact that the entire fly was used for RNAseq. Doing RNAseq on the animal as a whole will present an aggregate of overall gene expression changes throughout the body and will be dominated by tissues that are large or highly transcriptionally regulated upon infection. If the authors wish to understand the transcriptomics of behavioral modification – which is a main selling point of the manuscript – they would have been much better served to perform the host RNAseq on heads only. This is easy and routine in *Drosophila*. The attempt of the authors to dissect brains out of the head does not necessarily address this issue since gene expression could have changed drastically during the dissection procedure, which is much slower than decapitation, potentially leading to the non-conclusive results presented. The reviewers strongly recommend that the authors collect new transcriptomic data on heads only if they wish to draw conclusions about the gene expression determinants of manipulated behavior and not simply document standard transcriptional responses to infection. If the authors choose not to perform head-only RNAseq, the manuscript needs to be modified to indicate that the experiment performed is likely to be severely underpowered for detecting gene expression differences leading to manipulated behavior, probably explaining the absence of such signal in the data, and no further conclusion should be drawn.

The exclusion of two samples at 96 hours post-infection appears to be based in circular logic and is not well justified. The motivation for the transcriptomics is to identify genes that are differentially expressed after infection. However, two samples were excluded at 96 hours because they did not exhibit a host immune response (subsection “Transcriptional profiles of *E. muscae* 'Berkeley' and *D. melanogaster* over the course of infection”). It is not clear why the exclusion filter applied at 96 hours (subsection “Transcriptional profiles of *E. muscae* 'Berkeley' and *D. melanogaster* over the course of infection”) was not also applied at 24 and 72 hours. It seems inappropriate to use differential expression as an exploratory tool to characterize the response to infection and simultaneously as a diagnostic for infection. This is important because a major claim in the manuscript is that *E. muscae* infection elicits an immune response. Is there anything in the pattern of fungal transcripts to suggest these two samples are peculiar, which could support the exclusion of these samples? Why is the exclusion filter not also applied at 24 and 72 hours post-infection? This issue would all need to be thoroughly addressed in a revision, potentially including re-analysis and filtering of the transcriptome, and at least showing that the chosen approach does not affect the primary conclusions.

One animal has an aberrantly high expression of immune response genes, described in the Discussion section. The authors interpret this as likely to represent "immune-system overload". It is unclear what that refers to. The authors may be imagining something reminiscent of septic shock or cytokine storms in vertebrates? As far as we are aware, there is no analogous phenomenon in insects, and this "overload" seems highly speculative. A more likely explanation may be that this host has aberrantly high fungal titre, or unusual fungal gene expression. The titre should be possible to estimate at least coarsely from the transcriptomic data. The authors are "inclined" to interpret this individual as a "fluke" because it is an apparent outlier. But at the same time, it represents 17% of their data. When the total n is only 6, it is hard to identify whether even single unusual samples lie outside the biologically natural distribution.

The reviewers would also like to see the false positive gene expression of *E. muscae* reads in the Control samples. This to validate the expression patterns found for group I in Figure 4B. The first 48 hours of *E. muscae* transcriptomic data may not be meaningful since the% reads aligned is within the same ballpark as the rate of false positive mapping in the controls.

The authors map the fungal reads to a de-novo assembled transcriptome. However, they show capacity to culture the *Entomophthora* strain and harvest spores. Why not then sequence the DNA and generate a de novo genome? Even a draft genome may provide better reference for transcript mapping than the draft transcriptome does. However, the reviewers recognize potential complications with genome sequencing assembly, including potential ploidy issues, and we do not think that using a different fungal reference assembly would change any major biological conclusions. This comment is therefore a recommendation and not a requirement. In any case, the authors should provide stronger rationale for the approach they choose.

In the transcriptome assembly, was there any attempt to remove transcripts that are neither *Drosophila* nor *Entomophthora*? It appears that any transcript that did not map to the *Drosophila* genome was assumed to be *Entomophthora*, but it seems possible that there could be contamination by fungal associates of *Drosophila* in culture, potentially including *Saccharomyces* species. Can such reads be filtered out?

In subsection “Transcriptional profiles of *E. muscae* 'Berkeley' and *D. melanogaster* over the course of infection”, it is unclear how the authors know that the *E. muscae* transcriptome is contaminated by "some" fly transcripts. This would seem impossible if the *E. muscae* transcriptome is assembled from reads that did not match the fly transcriptome. A clarification in presentation may be warranted.

The methodological set-up with exposure to infective cadavers for 24 hours implies that samples which are collected 24 hours later may have experienced initial conidia landing and penetration 24-48 hours earlier. This may be an additional factor to discuss in subsection “The progression of *E. muscae* 'Berkeley' infection in lab-reared *D. melanogaster”*, when treating the variation in *E. muscae* progression and variation in RNAseq data among samples, because samples collected with 24 hour intervals may actually be closer to the previous or later sampling time point.

The reviewers have multiple suggestions for revising the writing and presentation of the manuscript, detailed as follows:

In some places, the language of the manuscript seems to overreach the data presented or potential generality of this system. For example, the last sentence of the abstract suggests that the *D. melanogaster* – *E. muscae* Berkeley system could broadly reveal how microbes manipulate host behavior. However, it is widely recognized that there are idiosyncrasies to specific host-pathogen interactions. For example, in this system, the authors describe that their *Entomophthora* strain invades the brain tissue. This is different from what has been found in e.g., *Ophiocordyceps* infections (Hughes et al., 2011, Fredericksen et al., 2017), and *Entomophthora* and trematode infections in ants (e.g., Loos-Frank and Zimmermann, 1976, Romig et al., 1980) in ants. Additionally, de Bekker et al., 2017 suggests that the molecular strategies that even highly related fungi use to manipulate behavior is not necessarily similar. Additionally, the authors suggest gravitaxis to be the determining factor for the summiting behavior rather than phototaxis. However, for baculovirus inducing tree top disease, for instance, phototaxis has been shown to be the determining factor for summiting. These differences are not fully discussed by the authors and could be expanded in the Discussion section. The statements made in the last sentence of the Abstract might be too general. The last sentences of the introductory paragraph of the Discussion section are more appropriate, stating that this work will explain us how this is done in *Entomophthora muscae* Berkeley-*Drosophila* interactions specifically. At the moment, we cannot state much more than this, especially since much of the current data is pointing towards high species specificity of these reactions (i.e., not even another *E. muscae* strain can be successfully employed to manipulate *Drosophila*, so mechanisms underlying this are likely very specific).

New experimental results appear in the Discussion section. These need to be moved to the Results section and treated appropriately. Specifically, in subsection “*E. muscae* in wild drosophilids”, it appears that the authors have applied their LSU and ITS genotyping to flies that were independently collected, including some that were collected prior to the collections that gave rise to the present manuscript. If this is the case, these data should be presented as results instead of being introduced in the Discussion section. In subsection “*E. muscae* in wild drosophilids”, the authors imply that they have infected house fly with their strain of *E. muscae*, but those experiments are not described in the Results section nor are they tagged with "data not shown" in the Discussion section. If this result is one that the authors would like to convey in the present manuscript, they should present it fully. In the Discussion section, the authors newly present RNAseq data from dissected brains, although they do not find any differentiation in gene expression. This section should be moved to the Results section, and a full description should be presented in the Materials and methods section, including how long it took to dissect the fly brains. Taking considerable time to perform dissections will change transcript levels, making the data a less accurate reflection of the gene expression levels at play. Thus, the conclusion that behavioral modification may be largely independent of transcriptional changes in the brain (subsection “Why is *E. muscae* 'Berkeley' in the brain?”) could be incorrect. Again, the reviewers strongly recommend repeating this analysis with whole fly heads that can be rapidly excised and will represent strong enrichment for the central nervous system.

The description in the Introduction of the *E. muscae* species complex and the host-specificity of fungi in this complex is somewhat confusing and could be more precise. It is evident that individual isolates or strains within this species complex show local adaptation to certain hosts (E.g. Gryganskyi et al., 2013, or Jensen et al., 2001), although whether these should be divided in separate species, separate morphs or something like "host-races" is unclear. The thorough description of the present isolate from drosophilid-flies, is therefore likely more or less specific to drosophilid flies, and the authors could be more precise about that in the Introduction, Results section and Discussion section. The authors discuss the uncertainty in the current knowledge of whether natural infections of other hosts occur in the wild (subsection “*E. muscae* in wild drosophilids”), but do not mention that local adaptation between different hosts has previously been consistently observed within the *E. muscae* species complex.

A general comment on the writing is that the results are presented as a somewhat colloquial descriptive narrative, which makes for an easy read but sometimes falsely conveys a lack of rigor and gives the impression of a commentary/descriptive natural history type of text rather than a scientific account. Quantitative results are presented in figures, although often without statistical contrast, but the text is almost completely non-numerical. There are many instances of this. One example is in subsection “Description of *E. muscae* 'Berkeley' infection in CantonS flies”, regarding behavior analysis of dying flies. There is a quantitative measure that is being made, and it is illustrated in Figure 2C, but lines of text could easily be interpreted to indicate that a researcher simply looked at the vials and made a qualitative assessment. Although an "analysis" is referred to in subsection “Description of *E. muscae* 'Berkeley' infection in CantonS flies”, that analysis is not described. Even in Figure 2C, there is no quantitative analysis or indication of variance in the behavioral measures despite more than 80 flies being individually measured. In another example in subsection “Description of *E. muscae* 'Berkeley' infection in CantonS flies”, "of note, flies housed in complete darkness are still observed to die in elevated positions", but nothing about this experiment is described. How many flies were observed? What proportion of them died in elevated positions? The reader cannot easily determine whether this is an anecdotal observation that may or may not have merit, versus a description of what happened in an overwhelming majority of a large sample size of flies that were systematically evaluated. The same critique can be made of the histological characterizations (e.g., subsection “*E. muscae* 'Berkeley' is present in the fly nervous system 48 hours after exposure”). Do described phenomena happen 100% of the time on exactly the same timing? 60-80% of the time within a 24 hour window? Quantitative measurements such as those in Figure 2—figure supplement 1 should also be described in the main text.

A number of possible mechanisms for fungal manipulation of fly behavior are proposed in the Discussion section but none of these are experimentally addressed so they remain quite speculative. It is notable that there is not a single reference to published literature in this passage, even though there is a body of literature out there. More comparison between the findings and suggestions of this work and previous work on other systems would be of value.

---

## [Author Response]

The reviewers strongly recommend that the authors collect new transcriptomic data on heads only if they wish to draw conclusions about the gene expression determinants of manipulated behavior and not simply document standard transcriptional responses to infection. If the authors choose not to perform head-only RNAseq, the manuscript needs to be modified to indicate that the experiment performed is likely to be severely underpowered for detecting gene expression differences leading to manipulated behavior, probably explaining the absence of such signal in the data, and no further conclusion should be drawn.

We agree that performing transcriptomics on the entire fly does is not the best way to approach identifying the genes underlying behavior. However, we did not conduct the infection time-course to gain insight into mechanism. Rather we carried out these experiments to examine the progression of the infection and presented it as such in the manuscript. We would love to have a better functional annotation of the *E. muscae* genome, but it has proven difficult to sequence (it is large – ~600 Mb, and repeat rich, making assembly very challenging) and is so distant from any well-characterized organism as to make functional annotation difficult.

We opted to sequence brains rather than heads to avoid sequencing transcripts arising from non-neural tissues, specifically the fat body, which is the predominant source of immune response transcripts. Studies that have looked at heads vs. brains see considerable gene expression contributions from the fat body in the head (Fujii and Amrein, 2002; Kadener et al., 2006). Previous work has shown high correlation of gene expression between heads and dissected brains, and that the quality of RNA from dissected brains is comparable to that from decapitated heads (Kadener et al., 2006). Gene expression in dissected brains is stable over the course of single-cell dissociation, a process that takes an order of magnitude more time than dissection (M. Rosbash, personal communication).

The exclusion of two samples at 96 hours post-infection appears to be based in circular logic and is not well justified. The motivation for the transcriptomics is to identify genes that are differentially expressed after infection. However, two samples were excluded at 96 hours because they did not exhibit a host immune response (subsection “Transcriptional profiles of E. muscae 'Berkeley' and D. melanogaster over the course of infection”). It is not clear why the exclusion filter applied at 96 hours (subsection “Transcriptional profiles of E. muscae 'Berkeley' and D. melanogaster over the course of infection”) was not also applied at 24 and 72 hours. It seems inappropriate to use differential expression as an exploratory tool to characterize the response to infection and simultaneously as a diagnostic for infection. This is important because a major claim in the manuscript is that E. muscae infection elicits an immune response. Is there anything in the pattern of fungal transcripts to suggest these two samples are peculiar, which could support the exclusion of these samples? Why is the exclusion filter not also applied at 24 and 72 hours post-infection? This issue would all need to be thoroughly addressed in a revision, potentially including re-analysis and filtering of the transcriptome, and at least showing that the chosen approach does not affect the primary conclusions.

The analysis presented showed gene expression changes in 24-72 hours and excluded allsamples at 96 hours, not just the two that showed aberrant immune responses. We apologize that this was not clearly conveyed. The samples were initially identified as potential outliers not just based on looking only at immune gene expression – hierarchical clustering (not shown in Figure 4) revealed that they were more similar in overall gene expression profile to control samples as opposed to other exposed flies. The discrepancy in the immune response was found after this and thought to perhaps explain why these flies looked more like controls than infected animals, perhaps indicating that they had been able to combat infection (which would be consistent with lower *E. muscae* titers than expected for these samples). Alternatively, it could be that they were simply slower to progress through the course of infection, which is why titers are low, though this explanation doesn’t reconcile why overall gene expression patterns look very similar to control animals. We performed this analysis on all living flies sampled at 24-96 hours and arrived the same conclusions as our 24-72 hour analysis.

During revision, we decided to completely re-analyze our raw data using a different pipeline that allowed us to account for every read in each sample. We present the results from this new analysis in Figure 4 and Figure 5, which, as per your comments, includes all living animals from 24-96 hour time points.

One animal has an aberrantly high expression of immune response genes, described in the Discussion section. The authors interpret this as likely to represent "immune-system overload". It is unclear what that refers to. The authors may be imagining something reminiscent of septic shock or cytokine storms in vertebrates? As far as we are aware, there is no analogous phenomenon in insects, and this "overload" seems highly speculative. A more likely explanation may be that this host has aberrantly high fungal titre, or unusual fungal gene expression. The titre should be possible to estimate at least coarsely from the transcriptomic data. The authors are "inclined" to interpret this individual as a "fluke" because it is an apparent outlier. But at the same time, it represents 17% of their data. When the total n is only 6, it is hard to identify whether even single unusual samples lie outside the biologically natural distribution.

This statement was made in direct reference to an assertion in Boosma, et al., 2014 that, in entomophthoralean fungi that proliferate protoplastically within the host, these cells go undetected by the host immune system until the final stage of infection, wherein the fungus puts on a cell wall in preparation of leaving the host and the host immune system is overwhelmed:

“In contrast, many entomophthoralean fungi proliferate by yeast-like protoplasts, i.e., single-cell structures without a cell wall (12, 21) (Figure 2B). The advantages of this cell form are that nutrient acquisition rates are probably higher and that fungal cells can multiply in the hemocoel without being detected by the insect immune system that uses cell wall epitopes as cues (21). However, producing infective conidia requires that clonal protoplasts switch to producing cell walls and hyphal bodies during the final stages of infection (13). When that happens, the host immune system is quickly overwhelmed by the massive presence of pathogen biomass. Entomophthoralean fungi have thus evolved a strategy to weaken host immune responses very different from that used by many hypocrealean fungi.”

We apologize that the wording of our statement was not clear and have reworded this sentence to better reflect the intended sentiment. In addition, we have removed the word “fluke” and framed our data as inconsistent with the overwhelmed immune system model, but not of sufficient power to formally preclude such an event.

The reviewers would also like to see the false positive gene expression of E. muscae reads in the Control samples. This to validate the expression patterns found for group I in Figure 4B. The first 48 hours of E. muscae transcriptomic data may not be meaningful since the% reads aligned is within the same ballpark as the rate of false positive mapping in the controls.

During revision, we decided to completely re-analyze our raw data using a different pipeline that allowed us to account for every read in each sample and showed that our initially-reported false positive rates had been significantly overestimated with Kallisto. False positive gene expression of *E. muscae* reads in control samples have been added to Figure 4.

The authors map the fungal reads to a de-novo assembled transcriptome. However, they show capacity to culture the Entomophthora strain and harvest spores. Why not then sequence the DNA and generate a *de novo* genome? Even a draft genome may provide better reference for transcript mapping than the draft transcriptome does. However, the reviewers recognize potential complications with genome sequencing assembly, including potential ploidy issues, and we do not think that using a different fungal reference assembly would change any major biological conclusions. This comment is therefore a recommendation and not a requirement. In any case, the authors should provide stronger rationale for the approach they choose.

We did sequence our isolate’s genome with this intention, but assembly progress was hampered by the enormous size (~600 Mb) and repeat content (>85%) of the genome. Due to the difficulty of genome assembly, we opted to generate and use a transcriptome assembly for our analysis. Though we now have a rather fragmented draft genome assembly (which we have now made public via NCBI, PRJNA479887), we were also of the opinion that alignment to our genome assembly wouldn’t change the major takeaways from the transcriptomic time course experiment. While revising this manuscript, we did go back and use our genome assembly as a secondary alignment. The number of reads that align to the genome annotation is after removing those that align to the transcriptome is approximately 10 percent of the number that aligned to the transcriptome, demonstrating that the transcriptome has fairly good coverage. The number of reads that align to the transcriptome and genome annotation are highly correlated, suggesting they are measuring reads from the same RNA source (the fungus) but giving complementary information. Adding the genome annotation to our pipeline identified *E. musace* ‘Berkeley’ as the source of many reads that didn’t previously align to any reference but did not change our overall results.

In the transcriptome assembly, was there any attempt to remove transcripts that are neither Drosophila nor Entomophthora? It appears that any transcript that did not map to the Drosophila genome was assumed to be Entomophthora, but it seems possible that there could be contamination by fungal associates of Drosophila in culture, potentially including Saccharomyces species. Can such reads be filtered out?

We put much effort towards removing transcripts that were of neither *D. melanogaster* nor *E. muscae* origin from our assembly, which is described in the Materials and methods section. Since this wasn’t clear, we have added a description of how our reference transcriptome was prepared to the main body of the text. Our new analysis pipeline explicitly removes reads that can be attributed to other sources including: bacteria and fungi found in our laboratory culture, which we identified by assembling transcriptomes after removing reads that aligned to either *D. melanogaster* and *E. muscae* and then included whole-genomes of identified microbial species in our alignment pipeline; viruses, which we identified with a similar analysis, and known sequencing contaminants like the coliphage phiX174 which is used to calibrate sequencers.

In subsection “Transcriptional profiles of E. muscae 'Berkeley' and D. melanogaster over the course of infection”, it is unclear how the authors know that the E. muscae transcriptome is contaminated by "some" fly transcripts. This would seem impossible if the E. muscae transcriptome is assembled from reads that did not match the fly transcriptome. A clarification in presentation may be warranted.

We present a new analysis of our transcriptomic data. This statement does not appear in our manuscript and so is no longer relevant.

The methodological set-up with exposure to infective cadavers for 24 hours implies that samples which are collected 24 hours later may have experienced initial conidia landing and penetration 24-48 hours earlier. This may be an additional factor to discuss in subsection “The progression of E. muscae 'Berkeley' infection in lab-reared D. melanogaster, when treating the variation in E. muscae progression and variation in RNAseq data among samples, because samples collected with 24 hour intervals may actually be closer to the previous or later sampling time point.

We’ve included a clause in subsection “The progression of *E. muscae* 'Berkeley' infection in lab-reared *D. melanogaster* to acknowledge that timing of infection is another source of variability in our data.

The reviewers have multiple suggestions for revising the writing and presentation of the manuscript, detailed as follows:In some places, the language of the manuscript seems to overreach the data presented or potential generality of this system. For example, the last sentence of the abstract suggests that the D. melanogaster – E. muscae Berkeley system could broadly reveal how microbes manipulate host behavior. However, it is widely recognized that there are idiosyncrasies to specific host-pathogen interactions. For example, in this system, the authors describe that their Entomopthora strain invades the brain tissue. This is different from what has been found in e.g., Ophiocordyceps infections (Hughes et al., 2011, Fredericksen et al., 2017), and Entomophthora and trematode infections in ants (e.g., Loos-Frank and Zimmermann, 1976, Romig et al., 1980) in ants. Additionally, de Bekker et al., 2017 suggests that the molecular strategies that even highly related fungi use to manipulate behavior is not necessarily similar. Additionally, the authors suggest gravitaxis to be the determining factor for the summiting behavior rather than phototaxis. However, for baculovirus inducing tree top disease, for instance, phototaxis has been shown to be the determining factor for summiting. These differences are not fully discussed by the authors and could be expanded in the Discussion section. The statements made in the last sentence of the Abstract might be too general. The last sentences of the introductory paragraph of the Discussion section are more appropriate, stating that this work will explain us how this is done in Entomophthora muscae Berkeley-Drosophila interactions specifically. At the moment, we cannot state much more than this, especially since much of the current data is pointing towards high species specificity of these reactions (i.e., not even another E. muscae strain can be successfully employed to manipulate Drosophila, so mechanisms underlying this are likely very specific).

We appreciate these suggestions and have modified the manuscript accordingly.

New experimental results appear in the Discussion section. These need to be moved to the Results section and treated appropriately. Specifically, in subsection “E. muscae in wild drosophilids”, it appears that the authors have applied their LSU and ITS genotyping to flies that were independently collected, including some that were collected prior to the collections that gave rise to the present manuscript. If this is the case, these data should be presented as results instead of being introduced in the Discussion section. In subsection “E. muscae in wild drosophilids”, the authors imply that they have infected house fly with their strain of E. muscae, but those experiments are not described in the Results section nor are they tagged with "data not shown" in the Discussion section. If this result is one that the authors would like to convey in the present manuscript, they should present it fully. In the Discussion section, the authors newly present RNAseq data from dissected brains, although they do not find any differentiation in gene expression. This section should be moved to the Results section, and a full description should be presented in the Materials and methods section, including how long it took to dissect the fly brains. Taking considerable time to perform dissections will change transcript levels, making the data a less accurate reflection of the gene expression levels at play. Thus, the conclusion that behavioral modification may be largely independent of transcriptional changes in the brain (subsection “Why is E. muscae 'Berkeley' in the brain?”) could be incorrect. Again, the reviewers strongly recommend repeating this analysis with whole fly heads that can be rapidly excised and will represent strong enrichment for the central nervous system.

We have included both the discussion of other observations of *E. muscae* in *Drosophila* (Supplementary file 1) and the brain expression data to the Results section (Figure 8) and have an expanded the methods section accordingly. We appreciate the reviewer’s concern about fly heads vs. dissected brains. We gave serious consideration to repeating this experiment in heads, but after examining the literature and discussing this with experts in gene expression in the fly nervous system, we do not think it is necessary to carry out what would be a time consuming and expensive experiment with limited hope of yielding valuable data.

As discussed above, we originally opted to sequence brains rather than heads to avoid sequencing transcripts arising from non-neural tissues, specifically the fat body, which is the predominant source of immune response transcripts. Studies that have looked at heads vs. brains see considerable gene expression contributions from the fat body in the head (Fujii and Amrein, 2002; Kadener et al., 2006). Previous work has shown high correlation of gene expression between heads and dissected brains, and that the quality of RNA from dissected brains is comparable to that from decapitated heads (Kadener et al., 2006). Gene expression in dissected brains is stable over the course of single-cell dissociation, a process that takes an order of magnitude more time than dissection (M. Rosbash, personal communication).

The description in the Introduction of the E. muscae species complex and the host-specificity of fungi in this complex is somewhat confusing and could be more precise. It is evident that individual isolates or strains within this species complex show local adaptation to certain hosts (E.g. Gryganskyi et al., 2013, or Jensen et al., 2001), although whether these should be divided in separate species, separate morphs or something like "host-races" is unclear. The thorough description of the present isolate from drosophilid-flies, is therefore likely more or less specific to drosophilid flies, and the authors could be more precise about that in the Introduction, Results section and Discussion section. The authors discuss the uncertainty in the current knowledge of whether natural infections of other hosts occur in the wild (subsection “E. muscae in wild drosophilids”), but do not mention that local adaptation between different hosts has previously been consistently observed within the E. muscae species complex.

We have addressed this issue in the Discussion section.

A general comment on the writing is that the results are presented as a somewhat colloquial descriptive narrative, which makes for an easy read but sometimes falsely conveys a lack of rigor and gives the impression of a commentary/descriptive natural history type of text rather than a scientific account. Quantitative results are presented in figures, although often without statistical contrast, but the text is almost completely non-numerical. There are many instances of this. One example is in subsection “Description of E. muscae 'Berkeley' infection in CantonS flies”, regarding behavior analysis of dying flies. There is a quantitative measure that is being made, and it is illustrated in Figure 2C, but lines of text could easily be interpreted to indicate that a researcher simply looked at the vials and made a qualitative assessment. Although an "analysis" is referred to in subsection “Description of E. muscae 'Berkeley' infection in CantonS flies”, that analysis is not described. Even in Figure 2C, there is no quantitative analysis or indication of variance in the behavioral measures despite more than 80 flies being individually measured. In another example in subsection “Description of E. muscae 'Berkeley' infection in CantonS flies”, "of note, flies housed in complete darkness are still observed to die in elevated positions", but nothing about this experiment is described. How many flies were observed? What proportion of them died in elevated positions? The reader cannot easily determine whether this is an anecdotal observation that may or may not have merit, versus a description of what happened in an overwhelming majority of a large sample size of flies that were systematically evaluated. The same critique can be made of the histological characterizations (e.g., subsection “E. muscae 'Berkeley' is present in the fly nervous system 48 hours after exposure”). Do described phenomena happen 100% of the time on exactly the same timing? 60-80% of the time within a 24 hour window? Quantitative measurements such as those in Figure 2—figure supplement 1 should also be described in the main text.

We agree with this observation and recommendation. We were trying to provide readers with a more accessible description of the *E. muscae* Berkeley infection cycle, but erred in making it too colloquial and anecdotal. We have rewritten that section to provide more precise, quantitative descriptions and thank the reviewer for helping us to improve the manuscript in this way.

A number of possible mechanisms for fungal manipulation of fly behavior are proposed in the Discussion section but none of these are experimentally addressed so they remain quite speculative. It is notable that there is not a single reference to published literature in this passage, even though there is a body of literature out there. More comparison between the findings and suggestions of this work and previous work on other systems would be of value.

We have modified the discussion to include examples from the literature for the models of behavior manipulation that we propose.